# Understanding the ability of low-cost MOx sensors to quantify ambient VOCs

Ashley M. Collier-Oxandale[1], Jacob Thorson[2], Hannah Halliday[3], Jana Milford[2], and Michael Hannigan[2]

[1]Environmental Engineering, University of Colorado Boulder, Boulder, 80309, USA
[2]Mechanincal Engineering, University of Colorado Boulder, Boulder, 80309, USA
[3]NASA Langley Research Center, Hampton, VA, 23666, USA

*Correspondence to*: Ashley M. Collier-Oxandale (ashley.collier@colorado.edu)

**Abstract.**

Volatile organic compounds (VOCs) present a unique challenge in air quality research given their importance to human and environmental health, and their complexity to monitor resulting from the number of possible sources and mixtures. New technologies, such as low-cost air quality sensors have the potential to support existing air quality measurement methods by providing high time and spatial resolution data. This higher resolution data could provide greater insight into specific events, sources, and local variability. Furthermore, given the potential for differences in selectivities for sensors, leveraging multiple

sensors in an array format may even be able to provide insight into which VOCs or types of VOCs are present. During the FRAPPE/DISCOVER-AQ monitoring campaigns, our team was able to co-locate two sensor systems, using metal oxide (MOx) VOC sensors, with a proton-transfer-reaction mass spectrometer (PTR-QMS) providing speciated VOC data. This dataset provided the opportunity to explore the ability of sensors to estimate specific VOCs and groups of VOCs in real-world conditions, e.g., dynamic temperature and humidity. Moreover, we were able to explore the impact of changing VOC

compositions on sensor performance as well as the difference in selectivities of sensors in order to consider how this could be utilized. From this analysis, it seems that systems using multiple VOC sensors are able to provide VOC estimates at ambient levels for specific VOCs or groups of VOCs, it also seems that this performance is fairly robust to changing VOC mixtures, and it was confirmed that there are consistent and useful differences in selectivities between the two MOx sensors studied. While this study was fairly limited in scope, the results suggest that there is the potential for low-cost VOC sensors to support

highly resolved, ambient hydrocarbon measurements. The availability of this technology could enhance research and monitoring for public health and communities impacted by air toxics, which in turn could support a better understanding of exposure and actions to reduce harmful exposure.

# 1 Introduction

## 1.1 Background

Volatile organic compounds (VOCs) are ubiquitous in daily life; from the naturally occurring scents of flowers blooming in the spring to VOCs resulting from human activity, such as BTEX emissions from vehicles, compounds emitted when cooking, and even fragrances in cleaning supplies and personal care products (McDonald et al., 2018). In addition to their ubiquitous nature, VOCs are wide ranging in terms of potential risks to our health. Many VOCs that pose a danger to human health are classified as Hazardous Air Pollutants (HAPs) by the US EPA (Woodruff et al., 1998). For example, two of the more toxic and prevalent compounds from the HAPs list are benzene and formaldehyde, both of which pose a variety of risks from acute toxic effects to long-term carcinogenic risks depending on the level of exposure (Suh et al., 200). The inhalation of benzene on shorter time scales can result in neurologic symptoms, such as dizziness, drowsiness, headaches, and unconsciousness (Suh et al., 2000). Formaldehyde has been cited as a concern for indoor air quality as it is a respiratory and sensory irritant (Rumchev et al., 2007). One study have found that children exposed to a median level of 20 µg/m$^3$ or more of benzene in their homes were eight times more likely to have asthma than children living in homes with lower levels of benzene; a similar link was found between formaldehyde and asthma with researchers finding a 3% increase in risk of having asthma for every 10 µg/m$^3$ increase in formaldehyde exposure (Rumchev et al., 2007). Additionally, researchers have found that both compounds rank among the highest in terms of cancer risk when personal exposure across microenvironments and different exposure pathways are taken into consideration (Loh et al., 2007).

Beyond these two compounds there are many other VOCs that warrant concern. The RIOPA Study measured VOC concentrations outside of homes to examine the impact of proximity to nearby emission sources on exposure. This study found certain VOC levels elevated 1.5-4 times above ambient levels for homes less than 50 meters from a source (Kwon et al., 2006). Among the VOCs studied during RIOPA were benzene and perchloroethylene, another known carcinogen. Researchers also found that living within 25 m of a source of either of these two compounds increased the lifetime upper-bound cancer risk by 50-200% as compared to living more than 250 m from a source (Kwon et al., 2006) – highlighting the importance of studying VOCs at increased spatial resolutions. Another study found that cancer risks as well as non-cancer neurological and respiratory risk benchmarks, as defined by the US EPA, were exceeded in two environmental justice communities for several compounds, including benzene (Wu et al., 2012). Beyond the health effects, VOCs are often the cause of odor complaints, such as those tied to industrial activity. Even if there are no confirmed health effects from a pollutant, exposure to odors can cause quality of life issues and have hidden societal costs such as stress-related physical disorders (Beloff et al., 2000).

Given the prevalence of hazardous VOCs, more measurements and data could help inform actions to reduce VOCs and the public's exposure. However, given the spatial and temporal variability for VOC sources and complex mixtures of VOCs that occur in the environment, new approaches may be needed to supplement existing methods. Currently there are a variety of methods to quantify ambient VOCs, including real-time instruments, whole air sampling techniques, and passive methods capable of providing accurate, speciated measurements (Krol et al., 2010; Kumar & Viden, 2007). However, relying on a

single high-quality instrument may miss important spatial patterns, and using a distributed method such as sorption tubes that provide time-averaged values may miss important temporal patterns. Next generation monitoring technologies, such as low-cost air quality sensors, used in combination with conventional techniques are an approach that may be able to help address these needs. Low-cost sensing systems often cost orders of magnitude less than conventional instruments on a per-unit basis

and are simpler to deploy and operate making them particularly well-suited to provide preliminary or supplementary data for community-based projects or projects in partnership with environmental justice communities where resources may be limited (Shamasunder et al., 2018). Deployments of these sensing systems have already demonstrated the capacity to provide information on pollutant variability at small scales (Cheadle et al., 2017; Sadighi et al., 2018; Collier-Oxandale et al., 2018a), to differentiate regional trends from local emissions (Heimann et al., 2015), and to support personal exposure monitoring

(Piedrahita et al., 2014; Jerrett et al., 2017). However, sensor performance in regards of quantification is an ongoing challenge for this technology. While some studies have demonstrated success quantifying sensors for gas-phase or criteria pollutants (Zimmerman et al., 2018; Casey et al., 2019; Cross et al., 2017), this task may be more complicated for VOC sensors given that ambient VOCs exist in complex and dynamic mixtures.

## 1.1 Previous VOC Sensor Research

One of the reasons quantification is a challenge for low-cost sensors is their cross-sensitivity to environmental factors, such as temperature and humidity, and also to confounding gases (Lewis et al., 2016). Addressing this issue of cross-sensitivity is more complicated for VOC sensors compared to single chemical sensors as there are many more confounding species to consider and calibration models will need to be trained to target specific VOCs or groups of VOCs (Lewis et al., 2016). Several reviews provide an overview of the different sensors and systems available for measuring VOCs, including information on

strengths and limitations, and performance evaluations based on laboratory tests and examples found in the literature (Spinelle et al., 2017a; Spinelle et al., 2017b; Szulcynski & Gebicki, 2017; Williams & Kaufamn, 2015). These reviews also discuss the cross-sensitivity and selectivity issue; when considering the potential for sensors to make ambient measurements of benzene, the reviews noted that most sensors lack the selectivity and sensitivity for these measurements when sensors were tested individually (Spinelle et al., 2017a; Spinelle et al., 2017b). Researchers have also confirmed via laboratory tests that the limit

of detection for most electrochemical and MOx sensors is too high for ambient measurements, and while photo ionization detector (PID) sensors are capable of lower detection limits with linear responses, these suffer from cross-sensitivities caused by interfering compounds. Similarly, laboratory evaluations conducted by the US EPA seemed to indicate there is the potential for these systems to record environmentally relevant levels of VOCs, however, they also found that only two of the five systems tested seemed capable of detecting VOCs below 25 ppb. While much of the current literature seems to suggest that the sensors

currently available either lack low detection limits or the selectivity to pick out the compounds of concern, there are examples of deployments and laboratory studies presenting some innovative techniques for using these sensors and analyzing the data that could yield the useful information.

For example, DeVito and colleagues (2008) applied a neural net calibration to an array consisting of five different MOx sensors and provided a relatively stable benzene prediction with less than 2% error, for a period of 6 months. Furthermore, this study was conducted at a stationary monitoring site near a road, where the sensor system was subject to ambient temperature and humidity variations as well as varied concentrations of other VOCs (De Vito et al., 2008). Speaking to the potential for improving sensor quantification, the results of this study meet the Data Quality Objectives (DQO) outlined by the European Air Quality Directive for indicative benzene measurements, which call for a relative error less than 30% (Spinelle et al., 2017a). An earlier study, Wolfrum and colleagues (2006) demonstrated the use of MOx sensors (the Figaro TGS 2602) in arrays to differentiate and quantify three different VOCs (toluene, acetone, and isopropanol) in a laboratory setting. In addition to detecting these compounds at sub-ppm levels (approximately 0.1 – 1 ppm), analysis confirmed the sensor array's ability to predict individual pollutants in the presence of a confounding VOCs further speaking to the potential for VOC sensors (Wolfrum et al., 2006). Another study by Eugster and Kling (2011), using a similar MOx sensor (the Figaro TGS 2600), demonstrated the detection of ambient methane in a remote area of Alaska throughout dynamic environmental conditions.

Other techniques being piloted to improve the capabilities of low-cost sensors include temperature-controlled operation (TCO) and/or utilizing a pre-concentrator (Schutze et al., 2017). TCO makes use of the fact that different gases react with the surface of the sensor at different temperatures as well as the dynamic response after cleaning the surface via heating to elevated temperatures (Schutze et al., 2017). In one such study, researchers found that by applying TCO to a sensor system with MOx sensors, they were able to achieve an accuracy of ±0.2 – 2 ppb depending on the target gas concentration (benzene) and level of confounding gas(es) (Saurwald et al., 2018). Another study, also using TCO and multiple MOx sensors, demonstrated the ability to differentiate VOCs (benzene, formaldehyde, and naphthalene) at the ppb level, even in the presence of a confounding gas (ethanol) at much higher concentrations than the target gases (Leidinger et al., 2014). This differentiation was achieved using Linear Discriminant Analysis and was able to correctly classify the gas 95-99% of the time for concentrations of 4.7, 100, and 20 ppb for benzene, formaldehyde, and naphthalene respectively (Leidinger et al., 2014). In addition to TCO, another technique under consideration is the addition of an open pre-concentrator where a target gas could accumulate on an adsorbing material and then be thermally desorbed for analysis (Schutze et al., 2017; Leidinger et al., 2016). A system such as this would facilitate lower detection limits for MOx sensor systems. In addition to different techniques for collecting and processing sensor data, another option is to combine low-cost sensors with other measurement techniques. For example, a study in Philadelphia involved combining low-cost sensors (in this case PID sensors) and passive adsorption tubes (Thoma et al., 2016). The passive adsorption tubes provided speciated, quantified VOC data, while the sensor data along with meteorological information provided valuable information regarding pollutant trends and emission sources (Thoma et al., 2016). While there are many challenges associated with the use of VOC sensors, the potential this technology has to complement current monitoring efforts necessitates the exploration of these innovative solutions.

During the FRAPPE/DISCOVER-AQ campaign in Colorado, our team placed two low-cost sensor systems at the Platteville Atmospheric Observatory (PAO), co-located with a proton-transfer-reaction mass spectrometer (PTR-QMS) that provided speciated VOC data. Each sensor system included two different MOx VOC sensors in addition to other gas-phase and

environmental sensors. It is this combination of the availability of speciated VOC data and the dynamic environmental conditions of a field deployment that make this dataset and the subsequent analysis unique. There are numerous studies exploring the performance of these types of sensors in the lab when exposed to different VOCs and even complex VOC mixtures. There are several field studies examining the deployment of these sensors, however, these studies tend to involve a single VOC reference instrument (e.g., benzene) or target VOC. This dataset will help to further inform best practices and procedures for using these sensors thanks to the added complexity of our reference data. In this paper we explore the quantification of these sensors for individual and grouped VOCs, we also examine the different selectivities of the two sensors to better understand how these differences can be leveraged, and finally we try to understand how consistent sensor performance might be across different atmospheric compositions. These results build off previous work that involved quantifying one of the MOx sensors for ambient levels of methane, allowing us to explore the advantages of multi-sensor systems (Collier-Oxandale et al., 2018b).

## 2. Methods

### 2.1 Deployment Overview

In the summer of 2014, during the FRAPPE and DISCOVER-AQ campaigns (Pfister et al., 2017), our team deployed a network of low-cost sensors systems in an attempt to quantify the small-scale spatial variability of pollutants. However, these measurement campaigns also provided the valuable opportunities to co-locate our systems with high quality, reliable reference instruments providing the opportunity to improve sensor quantification and validation. One such co-location was at the Platteville Atmospheric Observatory (PAO), where two sensor systems, termed U-Pods, were co-located with the NATIVE Trailer maintained by researchers from Penn State (Halliday et al., 2016). The two U-Pods were co-located on the roof of the NATIVE trailer for approximately one month from mid-July to mid-August, Figure 1c includes a photo of the two U-Pods. This site offered a unique dataset given the potential for different types of VOCs. The PAO is in a rural area to the northeast of populated urban areas and surrounded by nearby oil and gas activity; there was the potential for typical traffic and urban emissions as well as emissions from oil and gas activity, and possibly even from local agriculture. Figure 1a illustrates the site's placement in relation to nearby cities and active/inactive oil and gas wells.

In addition to the potential for a range of different VOCs, the potential for relatively high levels of VOCs also made the PAO a site well suited for exploring the capabilities of VOC sensors. As some researchers have observed, levels of VOCs in regions with heavy oil and gas development can be higher than in to typical urban environments (Helmig et.al., 2014). Informing our work, a study undertaken prior to the FRAPPE and DISCOVER-AQ campaigns found several species of alkanes and benzene to be on average higher in the rural oil and gas production area as compared to the nearby urban area of Denver (Thompson et al., 2014). In some instances, the dynamic ranges of VOCs in cities may be an order of magnitude less than those observed in near sources in oil and gas productions areas (Warneke et al., 2013; Warneke et al., 2014; Borbon et al., 2013). Therefore,

while enhanced levels of VOCs may support this initial field work, it is important to consider how sensor performance may vary in environments with different levels of VOCs or different VOC compositions.

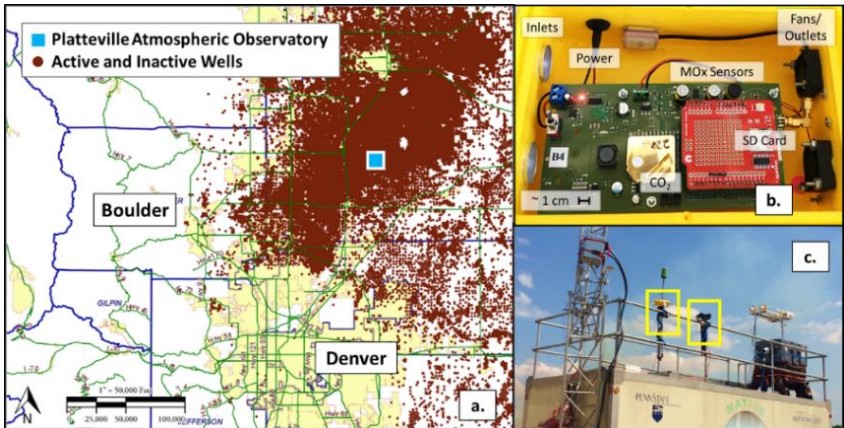

**Figure 1: The map (1a) illustrates the placement of the Platteville site with respect to the nearby cities Denver and Boulder, and oil and gas activity as indicated by active and inactive wells (COGCC, 2017. The top right photo (1b) shows the inside of a Y-Pod (a newer version of the U-Pod), which depicts the sensors used and the general design of the system. There were some updates to the circuit board, but the sensors used were the same. The photo on the bottom right (1c) shows the placement of the two U-Pods on the NATIVE Trailer at the Platteville site.**

This deployment is described in greater detail in our previous paper (Collier-Oxandale et al., 2018b). While VOC quantification was not the original intent of the deployment, it was something we were able to explore from the unique dataset provided through this co-location. Due to this analysis not being anticipated, techniques such as TCO were not incorporated; however, we did examine the use of a multi-sensor system in the context of relatively simple deployment and sensor performance quantification procedures. Thus, this work provides an opportunity to learn about VOC sensor potential at a fundamental level, under typical field conditions.

**2.2 Reference Measurements**

As previously mentioned, there were many reference instruments in operation at the PAO site, including a proton-transfer-reaction quadrupole mass spectrometer (PTR-QMS) that provided high-time resolution data for the VOC species listed in Table 1 (Halliday et al, 2016; Gouw & Warneke, 2007). Routine operating parameters and procedures were applied and the PTR-QMS was run in multiple ion detection mode (Halliday et al., 2016). Halliday and colleagues provide a detailed description of the operation of the PTR-QMS during this campaign as well as more information on the other measurements occurring at the NATIVE Trailer in a study examining ambient benzene ( 2016). The analysis presented here also relies on data from a Los Gatos instrument (model number 911-001), which utilizes cavity-enhanced absorption spectroscopy (CEAS) in which the optical cavity creates an effective pathlength several kilometres long resulting in improved sensitivity (O'Shea et al., 2013). This instrument was also maintained and operated by the NATIVE Trailer team. In addition to the speciated VOC

and methane data, the Penn State NATIVE Trailer was outfitted to measure the other pollutants listed in Table 1. All the reference data was retrieved from the Discover-AQ data repository (NASA, 2015).

**Table 1: Reference data utilized in this analysis and the instruments used to collect those data**

| Reference Pollutant | Instrumentation | Reference Pollutant | Instrumentation |
|---|---|---|---|
| Acetaldehyde | PTR-QMS | Methane ($CH_4$) | Los Gatos CEAS |
| Acetone-Propanol | PTR-QMS | Carbon Monoxide (CO) | Thermo CO Analyzer |
| Benzene | PTR-QMS | Carbon Dioxide ($CO_2$) | Licor 7000 |
| $C_8$ Alkylbenzenes | PTR-QMS | Nitric Oxide (NO) | Thermo $NO_y$ Analyzer |
| $C_9$ Alkylbenzenes | PTR-QMS | Nitrogen Dioxide ($NO_2$) | Environment SA Analyzer |
| Formaldehyde | PTR-QMS | Ozone ($O_3$) | Thermo $O_3$ Analyzer |
| Methanol | PTR-QMS | Hydrogen Sulfide ($H_2S$) | PTR-QMS |
| Toluene | PTR-QMS | | |

## 2.3 MOx Sensors and the U-Pod Platform

MOx sensors are composed of a metal-oxide surface (often tin dioxide), a sensing chip to measure changes in conductivity, and a heater. The general mechanism is that oxygen molecules adsorb to the metal oxide surface, trapping electrons. When the sensor comes in to contact with the target reducing gas these oxygen molecules react and are removed allowing the electrons to flow and increasing the conductivity across the surface (Wang et al., 2010). While the principle is simple, complications arise during quantification of pollutant concentrations, as the reactions occurring on the sensor surface are impacted by changes
in temperature and humidity (Wang et al., 2010, Sun et al., 2012) as well as the fact that these sensors are cross-sensitive to gases other than the target gas (Spinelle et al., 2017b). Adding to this complexity, the nanostructure of the metal-oxide surface itself can also influence sensitivity and selectivity (Sun et al., 2012; Shen et al., 2018). These sensors were developed for and are typically used in scenarios where high pollutant concentrations would be expected, such as in an industrial setting or inside a vehicle engine.

Two VOC sensors are incorporated into our sensing platform, the TGS 2600 and the TGS 2602 (Figaro, Inc.). They are advertised for the detection of "air contaminants"; the manufacturer specifies a few different contaminants to which they are sensitive. These include methane, carbon monoxide, iso-butane, ethanol, and hydrogen for the TGS 2600 (Figaro, 2005a). Hydrogen, ammonia, ethanol, hydrogen sulphide, and toluene sensitives are specified for the TGS 2602 (Figaro, 2005b). Granted several of these contaminants are not VOCs, however, as we are utilizing these sensors for the detection of VOCs we
will continue to identify them as VOC sensors. Additionally, both datasheets list a typical detection range of approximately 1-30 ppm (Figaro, 2005a & 2005b). However, as indicated by previous studies, sub-ppm levels of detection appear possible for both the TGS 2600 (Eugster & Kling, 2012) and the TGS 2602 (Wolfrum et al., 2006). The other environmental and gas-phase sensors used in the U-Pod are listed in Table 2 for reference.

The U-Pod is an embedded sensor system based on an open-source design developed and assembled by our lab (Mobile Sensing
Technology, 2017). These systems are housed in a small weather proof case, approximately 20cm x 25cm x 10cm, and use fans to pull air over the sensors and facilitate active flow. The U-Pods draw roughly 11 Watts of power and were powered by

12V AC/DC power adapters for this deployment, however they are capable of being powered by car batteries and/or solar power if remote deployment is necessary. The data is logged to an onboard micro-SD card at a rate of one data point every 6-25 seconds, depending on how the system is programmed. Figure 1b includes a diagram of the interior or a Y-Pod, a newer version of the technology utilizing the same sensors. U-Pods and newer versions of the system (the Y-Pods) have been used in several indoor and outdoor air quality studies that included sensor quantification and an examination of spatial variability or air quality trends Cheadle et al., 2017; Sadighi et al., 2018; Collier-Oxandale et al., 2018a).

**Table 2: Complete list of sensors used in the U-Pod**

| Sensor Type | U-Pod |
| --- | --- |
| Temperature & Relative Humidity | RHT03 (aka DHT22) |
| Temperature & Pressure | 47 Bosh BMP085 |
| Carbon dioxide | ELT S-100 NDIR |
| Ozone | SGX Corporation MiCS-2611 |
| VOC Sensor 1 | Figaro TGS 2600 MOx |
| VOC Sensor 2 | Figaro TGS 2602 MOx |
| Additional Optional Sensors | Alphasense B4 series (CO, NO, $NO_2$, $O_3$, $SO_2$), Baseline Mocon PID |

### 2.4 Data Processing and Analysis Rationale

The variable voltage values associated with the changing conductivity of the sensors are recorded to a micro-SD card and this voltage is then converted to a normalized resistance value ($R_s/R_0$), which is typically the form used for sensor data analysis (Eugster & Kling, 2012; Piedrahita et al., 2014). The resistance is first calculated using Eq. 1, provided by the sensor manufacturer. In this equation Rs is the changing resistance in the sensor driven by the concentration of the target gas, while $V_c$ is the circuit voltage, $R_l$ is the load resistance, and $V_{out}$ is the logged voltage. The $R_0$ value is typically the sensor resistance in clean air, and this value is used to normalize the resistance values. When calibrating in the field $R_0$ is identified as the maximum resistance value for the training period, or when the air is cleanest. For the following analysis this normalized term, $R_s/R_0$, is used. During processing, minute-medians are calculated from the sub-minute raw data, and warm-up data (the first half hour of operation after a U-Pod has been powered off for any period over half an hour) is removed.

$$R_S = \frac{V_c * R_L}{V_{out}} - R_L \qquad (1)$$

To facilitate analysis, the minute-median data was matched to the reference data using the nearest minute. This matched sensor (Table 2) and reference data (Table 1) was then five-minute averaged in blocks, in order to reduce the potential for any lags resulting from issues with time matching the data, particularly between short term spikes. Additionally, if three or more minutes were missing from either the sensor dataset or the reference dataset, then the whole five-minute average was excluded from the analysis. Occasional gaps in the reference datasets last from a few minutes to a few hours and vary by instrument; missing data was typically due to calibration events. One of the U-Pods, identified as P1 for this study, experienced a power failure resulting in approximately three days of data loss; the remaining data is complete. The second U-Pod, identified as P2 in this analysis, experienced intermittent power failure resulting in approximately 12 days of data lost in total out of the 22 days of

deployment. However, the sensors remained fully functional throughout the deployment, despite power failures. While this analysis primarily utilizes data from U-Pod P1, the data from P2 still provides an opportunity to validate our observations drawn from the P1 analysis.

To assess a VOC sensor's capabilities for use in the field, we applied typical quantification and analysis techniques. Given the cross-sensitives previously mentioned, researchers have found 'field calibration' or 'field normalization' to be a promising method to mitigate cross-sensitivities and calibrate for a target pollutant. Field calibrations are implemented by co-locating low-cost sensor systems with high-quality reference instruments (typically regulatory-grade, Federal Reference Method/Federal Equivalence Method monitors), often before and after a field deployment, and then generating a calibration model using an approach such as multiple linear regression or machine learning (Sadighi et al., 2018; Zimmerman et al., 2018; Cross et al., 2017). This technique allows predictive calibration models to be built for the conditions which sensors will experience in the field, such as diurnal environmental trends and background pollutants. While laboratory studies are valuable for understanding sensor capabilities and limitations in a controlled environment, researchers have continually observed that field as opposed to lab calibrations provide better pollutant estimations (Piedrahita et al., 2014; Castell et al., 2017). Therefore, the co-located data from the PAO site was used to conduct a typical field calibration by selecting a portion of the data from the beginning and the end of the deployment to build calibration models. The remaining data was used as testing data.

The models selected utilize multiple linear regression (MLR). While more complex machine learning techniques have proved very successful (Zimmerman et al., 2018; Casey et al., 2019; De Vito et al., 2008), we wanted to start with a simpler quantification method that is more easily understood and interpreted. Models were trained to estimate benzene, summed aromatic species, summed total VOC species, and methane. For summed signals, the ppbC values for each compound were calculated as the number of carbons in the compound multiplied by the volumetric concentration of the same compound; these ppbC values were then summed (Chen et al., 2014). The summed aromatics signal was calculated as the sum of the ppbC concentration values for benzene, $C_8$ alkylbenzenes, $C_9$ alkylbenzenes, and toluene. The total VOC species signal was calculated by summing the ppbC values for all of the available species measured by the PTR-QMS: acetaldehyde, acetone, benzene, $C_8$ alkylbenzenes, $C_9$ alkylbenzenes, formaldehyde, methanol, and toluene. The unit ppbC was selected as the value to be summed as it was thought that this unit would provide a more meaningful summed signal. The sum of ppbC values takes into account some of the differences between the individual compounds contributing to this sum. While this method does weight our signal for larger compounds this seems a reasonable approach as we also expect the size of the compound to factor into the magnitude of sensor response since we are measuring changes based on a chemical reaction on the sensor's surface.

Table 3 lists the multiple linear regression models utilized. Model 1 is a simple model including the two MOx VOC sensors, an interaction between the two sensors, and environmental predictors (e.g., temperature and humidity), and time to address drift. While Model 1 is the same for each target pollutant, Model 2 is different for each target group. For each Model 2, predictors were added to improve the resulting statistics and residuals for a given target pollutant or pollutant group. In addition to simulating a field calibration, we also utilized bootstrapping and analysis of variance to get a more fundamental understanding of the sensors' selectivities and the consistency of their behavior. Model 1 was selected for this analysis as it is

similar to multiple linear regression models typically used when exploring sensor performance (Spinelle et al., 2015; Zimmerman et al., 2018; Casey et al., 2019). The additional predictors added to Model 2, determined through trial and error, facilitate a look into whether there might be potential to improve these models by addressing the patterns in the residuals.

**Table 3: Multiple linear regression models used**

| Model Identifier | Model |
|---|---|
| Model 1: for all | $C = p_1 + p_2 * VOC_1 + p_3 * VOC_2 + p_4 * (VOC_1 * VOC_2) + p_5 * Temp. + p_6 * Abs.Hum. + p_7 * Time$ |
| Model 2: for benzene | $C = p_1 + p_2 * VOC_1 + p_3 * VOC_2 + p_4 * (VOC_1 * VOC_2) + p_5 * Temp. + p_6 * Abs.Hum. + p_7 * Time$ $+ p_8 * VOC_2(V) + p_9 * (Temp. * VOC_2)$ |
| Model 2: for aromatics | $C = p_1 + p_2 * VOC_1 + p_3 * VOC_2 + p_4 * (VOC_1 * VOC_2) + p_5 * Temp. + p_6 * Abs.Hum. + p_7 * Time$ $+ p_8 * VOC_2(V) + p_9 * (Temp. * VOC_2)$ |
| Model 2: for VOCs | $C = p_1 + p_2 * VOC_1 + p_3 * VOC_2 + p_4 * (VOC_1 * VOC_2) + p_5 * Temp. + p_6 * Abs.Hum. + p_7 * Time$ $+ p_8 * (\ln(Temp.) * Abs.Hum.)$ |
| Model 2: for methane | $C = p_1 + p_2 * VOC_1 + p_3 * VOC_2 + p_4 * (VOC_1 * VOC_2) + p_5 * Temp. + p_6 * Abs.Hum. + p_7 * Time$ $+ p_8 * CO2$ |

**Model predictors: $VOC_1$ – Figaro 2600 R/R0, $VOC_2$ – Figaro 2602 R/$R_0$, Temp – temperature (degrees C), Abs. Hum. – absolute humidity, Time – continuous time, $VOC_2(V)$ – Figaro 2602 voltage signal, $CO_2$ – carbon dioxide concentration (in this case from the reference data); C is the concentration value being solved for, either an individual or group of VOCs**

Clarifying how target VOCs and groups of VOCs were selected: benzene and summed aromatic species (including $C_8$ and $C_9$ alkylbenzenes) were selected for health reasons, as discussed in Section 1. While benzene health risks are the most well understood, the other common aromatic species (e.g., the BTEX compounds: benzene, ethylbenzene, toluene, and xylene) also present similar concerns for human health (Adgate et al., 2014). The summed total VOC signal was selected to provide some insight into the sensors' capacity to predict total non-methane organic compounds (TNMOCs) or possibly total non-methane hydrocarbons (TNMHCs). This signal was calculated by summing the ppbC values for all of the available species measured by the PTR-QMS: acetaldehyde, acetone, benzene, $C_8$ alkylbenzenes, $C_9$ alkylbenzenes, formaldehyde, methanol, and toluene. This type of measurement may be useful in an area concerned with a broad array of air toxics or when used in combination with a method of VOC speciation.

## 3. Results & Discussion

### 3.1 Field Calibration Performance

In the following sections, we show the results of the two MLR models for predicting each of the target VOCs or groups of VOCs. In each case, a timeseries is included to illustrate qualitatively the models' ability to predict trends and VOC concentrations, while regression statistics note the performance of the model across training and testing data and any changes from Model 1 to Model 2. The training periods have been highlighted in yellow and are the same training and testing periods used in the previous methane quantification work (Collier-Oxandale, 2018b). Also included are scatterplots to highlight

improvements in testing data from Model 1 to Model 2, and boxplots of the residuals (observed – predicted) to show where the majority fall (despite the wide variance apparent in the scatterplot). Additional residual plots are available in the supplemental (Figures S2)

### 3.1.1 Estimating Benzene and Summed Aromatic Species

Figures 2 and 3 present the results for benzene and the summed aromatic level quantification. Overall, both models capture the diurnal trends and short term elevated concentrations, although the models under predict the highest concentration events. Model 2 performs better than Model 1, with $R^2$ values of 0.67 and 0.64 for benzene and summed aromatics respectively. In both cases, Model 2 pulls some of the more extreme values closer to the 1:1 line. For the aromatics, Model 2 also results in closer fitting values at low concentrations. Furthermore, the RMSE values for the Model 2 testing data, 0.52 ppb and 11.25

ppbC for benzene and summed aromatics respectively, are less than the dynamic range observed in this dataset suggesting estimations can be made at the ambient levels observed during this deployment. The underprediction of the benzene and summed aromatic peaks is most likely due to a limitation associated with sensor response time. The response time for the PTR-QMS has a 1 second per species integration time during the 1-minute measurement cycle (Gouw & Warneke, 2007), however the MOx sensors respond more slowly as they are relying not only on a chemical reaction on the surface of the sensor, but also

the diffusion of the target gas to that surface. This means that a sensor may not be able to reach steady state in the time it takes for a plume to pass.

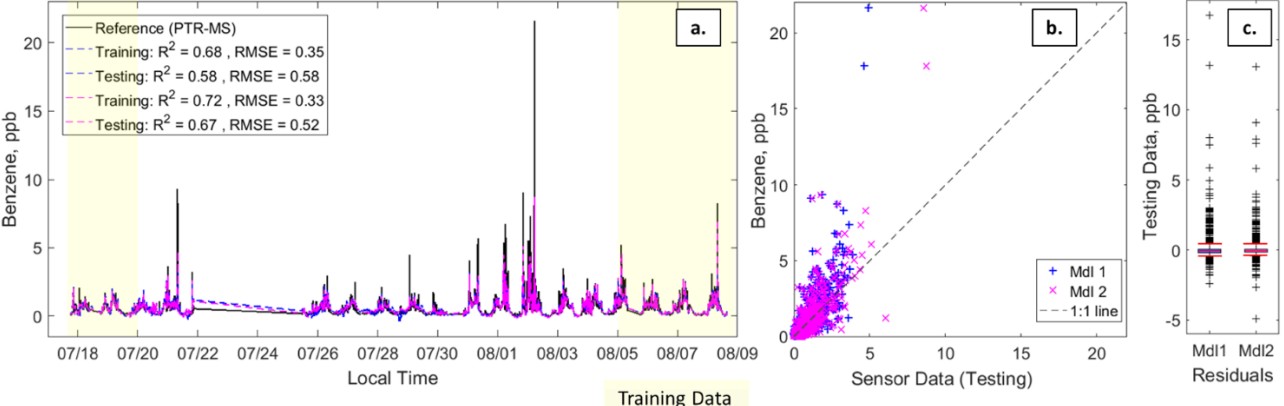

**Figure 2: The far-left panel (2a) depicts a time series including the benzene data from the PTR-QMS, and the fitted sensor data from Model 1 (in blue) and Model 2 (in fuchsia). The text box includes the following statistics: coefficient of determination ($R^2$) and the**

**root mean squared error (RMSE) in that order for each testing and training dataset. The middle panel (2b) depicts a scatterplot of the testing data for Model 1 (in blue) and Model 2 (in fuchsia), the 1:1 line has also been added. The furthest right panel (2c) depicts boxplots of the residuals with the whiskers at the 5th and 95th percentile respectively (in red).**

Despite this limitation, these two figures seem to suggest that these sensors can provide real-time estimates on aromatics and possibly even BTEX level estimates, a measure that could be especially valuable for exposure and health studies.

Given that ethylbenzene and xylenes are $C_8$ alkylbenzenes, the relatively strong performance of the summed aromatics prediction supports the idea that these sensors are suited for a more targeted BTEX concentration estimates. Additionally, it is possible that more advanced analytical techniques, such as neural networks, could find better preforming models, as shown in the work of De Vito and colleagues with MOx sensors and benzene (2008).

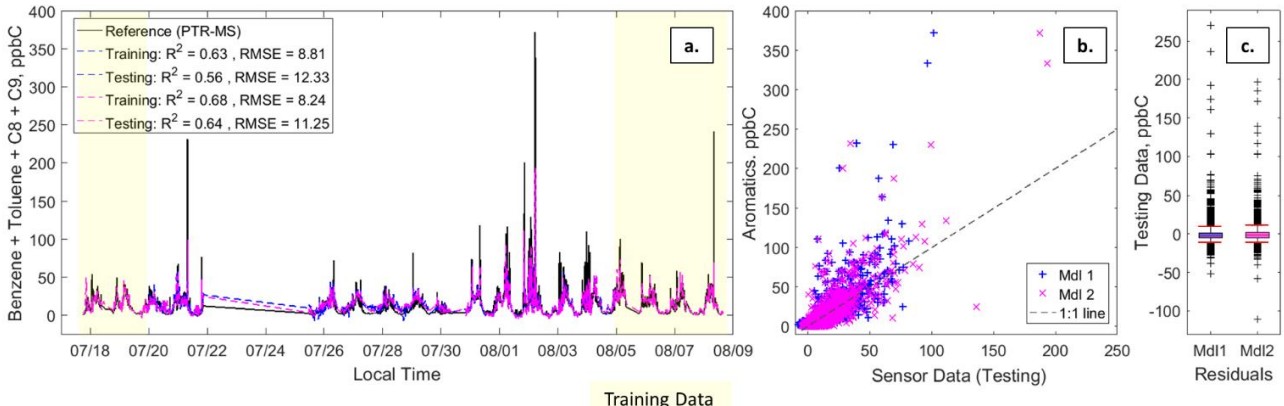

**Figure 3: The far-left panel (2a) depicts a time series including the summed aromatic data from the PTR-QMS (benzene, toluene, $C_8$ and $C_9$ alkylbenzenes), and the fitted sensor data from Model 1 (in blue) and Model 2 (in fuchsia). The text box includes the following statistics: coefficient of determination ($R^2$) and the root mean squared error (RMSE) in that order for each testing and training dataset. The middle panel (2b) depicts a scatterplot of the testing data for Model 1 (in blue) and Model 2 (in fuchsia), the 1:1 line has also been added. The furthest right panel (2c) depicts boxplots of the residuals with the whiskers at the 5th and 95th percentile respectively (in red).**

### 3.1.2 Estimating Summed VOCs

Figure 4 illustrates the performance of both models for the sum of all VOC compounds available from the PTR-QMS. Again, there are improvements with the more specialized Model 2, which corrects for some of the over-predictions. The resulting RMSE values of 13.38 and 12.78 ppbC and reasonably high $R^2$ values of 0.59 and 0.62 for Models 1 and 2 respectively, suggest this performance is suitable for certain ambient studies as the uncertainty is again well below the observed dynamic range. This analysis also demonstrates the stability of these models, as the signal being predicted is a sum whose precise composition is varying in time. For instance, the PTR-QMS signals from the aromatic species are well-correlated with each other (Figure S1), but the aromatics are not well-correlated with any of the OVOCs, meaning the relative amounts of these compounds vary, potentially making the task of signal prediction more challenging. These results may also support the assertion that these sensors could be suited to support TNMOC or TNMHC measurements, a measurement sometimes made in areas where VOCs are a concern and real-time data is desired. For example, a network of these sensors might be able to provide additional information on the variability or transport of TNMOCs or TNMHCs when used in conjunction with a higher quality instrument. Though speciated VOC measurements, made using canisters or field gas chromatographs for example, are also used. In terms of limitations, the models again under-predict the short-term peaks and this summed VOC estimate is noisier than for the previous two target VOCs (benzene and summed aromatics).

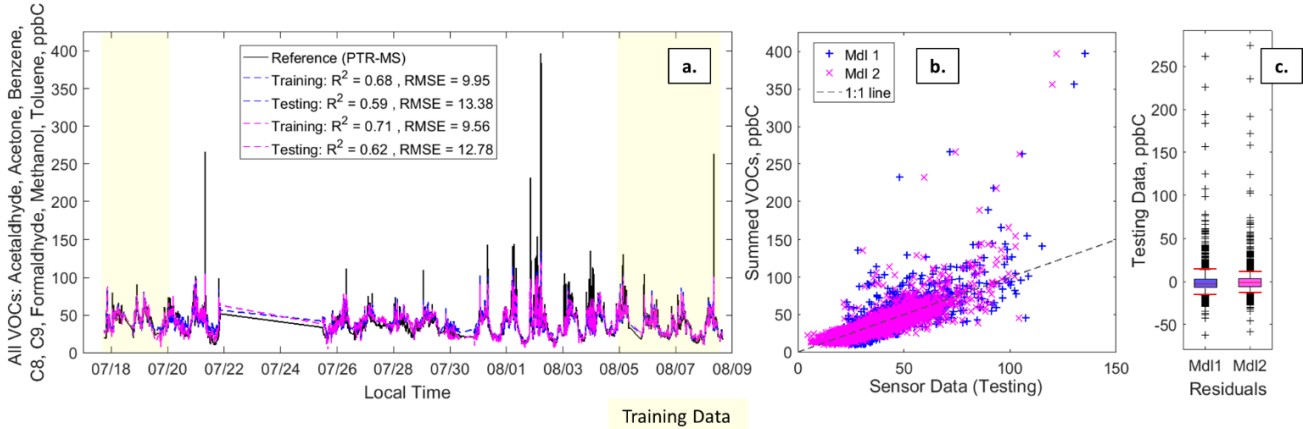

**Figure 4: The far-left panel (2a) depicts a time series including the summed VOC data from the PTR-QMS (acetaldehyde, acetone, formaldehyde, methanol, and the aromatic species), and the fitted sensor data from Model 1 (in blue) and Model 2 (in fuchsia). The text box includes the following statistics: coefficient of determination ($R^2$) and the root mean squared error (RMSE) in that order for each testing and training dataset. The middle panel (2b) depicts a scatterplot of the testing data for Model 1 (in blue) and Model 2 (in fuchsia), the 1:1 line has also been added. The furthest right panel (2c) depicts boxplots of the residuals with the whiskers at the 5th and 95th percentile respectively (in red).**

### 3.1.3 Estimating Methane

Figure 5 depicts the performance of the models for methane quantification. As with the previous VOCs and groups of VOCs, the methane calibration models are able to predict periods of elevated methane and indicate some of the shorter-term methane plumes. There are also the same limitations as noted in the previous section, mainly the under-prediction of peaks. Expanding on our previous methane quantification work, which utilized only the Figaro TGS 2600 sensor, including the second VOC sensor improves our ability to predict methane levels. The $R^2$ and RMSE for the testing data in this previous work was 0.50 and 0.383 ppm respectively (Collier-Oxandale et al., 2018b). Simply adding an additional sensor to the model results in an $R^2$ and RMSE of 0.58 and 0.24 ppm for the Model 1 testing data. Model 2 is even better performing. While the inclusion of a second VOC sensor does help to better target and predict methane, there is still room for improvement as is evidenced by the curvature in Figure 5b. As previously noted, it is possible that non-linear models or the use of a more advanced machine learning technique could facilitate further improvements. However, the overall results indicate that the use of multiple gas sensors does help mitigate the cross-sensitivities noted in the previous paper and improve the performance of the calibration model (Collier-Oxandale et al., 2018b).

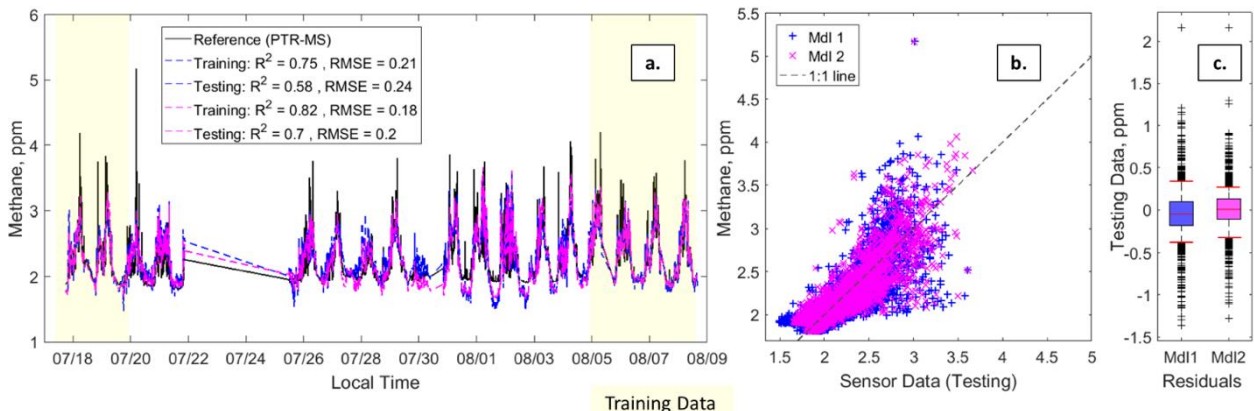

**Figure 5: The far-left panel (2a) depicts a time series including the methane data from the Los Gatos cavity ring-down instrument, and the fitted sensor data from Model 1 (in blue) and Model 2 (in fuchsia). The text box includes the following statistics: coefficient of determination ($R^2$) and the root mean squared error (RMSE) in that order for each testing and training dataset. The middle panel (2b) depicts a scatterplot of the testing data for Model 1 (in blue) and Model 2 (in fuchsia), the 1:1 line has also been added. The furthest right panel (2c) depicts boxplots of the residuals with the whiskers at the 5[th] and 95[th] percentile respectively (in red).**

### 3.1.4 Model Accuracy, Specialization, and Robustness

The cumulative distributions of the relative error (in percent) for each set of fitted data (including the testing period only) are shown in Figure 6. Considering the relatively simple deployment and quantification procedures, this figure emphasizes the utility and potential for these MOx sensors. Applying the 30% relative error DQO for indicative benzene measurements required by the European Air Quality Directive to our measurements, upwards of 98% and 84% of the methane and summed VOC estimates meet this benchmark (Spinelle et al., 2017a). For benzene and the summed aromatics, this number is lower, 43% and 38% respectively. However, these larger relative errors seem to be primarily driven by fairly small differences in low observed concentrations and the associated predictions. For example, a 100% relative error resulting from an observed value of .5 ppb and a predicted value of 1 ppb. If low values are excluded from the datasets (instances where the reference data is below 0.5 ppb for benzene and below 10 ppbC for the summed aromatics, roughly the RMSE for each dataset), then the proportion of data meeting the benchmark increases to 67% for benzene and 63% for summed aromatics. Excluding these low values might be reasonable for a study using sensors as the higher concentrations are most likely what would be of interest. Although it is important to reiterate that it's likely different ranges and compositions of VOCs will be encountered in different environments. To better understand what sensor performance may look like at lower concentrations, the regression analysis from Section 3.1.1 was repeated for benzene using only values less than 0.75 ppbV. The upper limit was selected as the range of benzene observed during the CalNex campaign in Los Angeles found benzene to be typically between ~ 0.1 - 0.6 ppbV (Warneke et al., 2013, Borbon et al., 2013). The resulting plots are available in the supplemental (Figure S4). For this new analysis, the R-squared values for the testing data remain fairly high at 0.68 for both Models 1 and 2, and the RMSE is 0.12 ppb, which is lower than the new dynamic range. While it's possible this consistent performance across different ranges of benzene is helped by well-correlated aromatic compounds that are essentially enhancing the trend for the sensor to detect; it

may also suggest that it is possible to train these calibration models for detection at lower levels. It would be helpful for future studies to more closely examine the lower detection limits and accuracy of these types of sensors for low levels of VOCs in

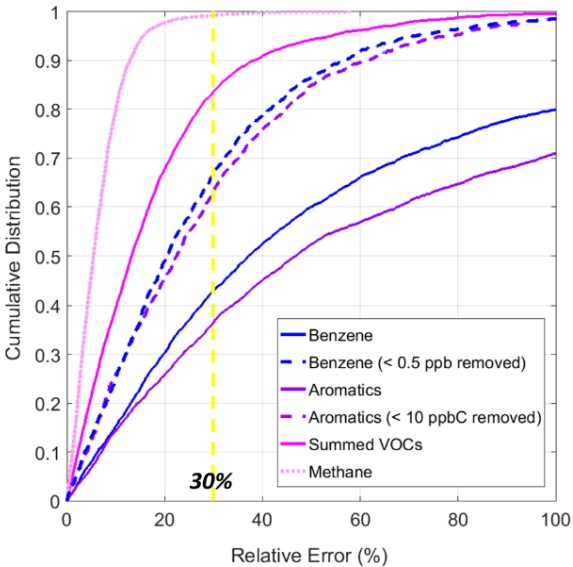

the field.

**Figure 6: Cumulative distributions of relative sensor error for all fitted testing or validation data, the two dotted lines represent the fitted testing data for benzene and summed aromatic, respectively, with the low values removed from the reference data.**

Given the limitations caused by cross-sensitivities, understanding how consistently sensors perform across changing compositions of VOCs is of high importance. While laboratory studies have illustrated the potential to identify specific pollutants in the presence of known confounders (Leidinger et al., 2014; Leidinger et al., 2017), the complex nature of field data requires a different approach. In Figure 7, the residuals from each Model 2 are plotted against other VOCs. Figure 7a depicts the residuals for the benzene model versus the ppbC sum of all remaining non-methane VOCs. Figure 7b depicts the summed aromatic residuals versus the ppbC sum of all non-methane and non-aromatic species. Figure 7c depicts the summed VOC residuals versus methane, and in Figure 7d the methane residuals are plotted versus the ppbC sum of all non-methane VOCs. For each plot, two concentrations, indicated by colored markers, were selected to be held constant; these are roughly the 75th and 95th percentile values for the predicted datasets. The intention of this analysis was to examine the effects of varying levels of VOCs on model performance, and the reason for selecting the 75th and 95th percentile values was based on an interest in measuring elevations in VOCs. The strength of low-cost VOC sensors, particularly as approaches to improve sensitivity are developed, will likely be in monitoring for relatively large elevations above background, therefore this is the application being considering.

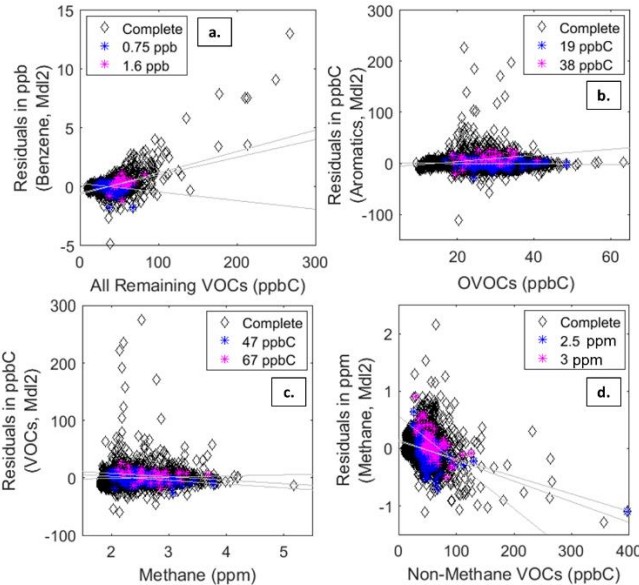

**Figure 7: Complete residuals verses other, non-target VOCs. The complete residuals for the Model 2 results are in black. The blue and fuchsia points that represent two values held constant. For panel a, the constant benzene values selected were 0.75 and 1.6 ± 0.05 ppb. For panel b, the summed aromatics values selected were 19 and 38 ± 1 ppbC. For panel c, the summed VOC values selected were 47 and 67 ± 1 ppbC. For panel d, the methane values selected were 2.5 and 3 ± 0.05 ppm.**

Figure 7b and 7c suggest that the predictions are robust across changing concentrations of other VOCs, as there are no positive or negative trends. In Figure 7c, the residuals seem to be largest for lower levels of methane and smaller for high levels of methane, which is possibly suggesting that summed VOCs are easier to estimate in the presence of higher concentrations of methane. However, this pattern is less apparent when we hold the estimated VOC concentrations constant. The negative and positive trends in Figures 7a and 7d emphasize that there is room for improvement in the models that predict a single compound. In Figure 7a, the positive trends once again highlight under-prediction of the highest benzene elevations. Given the correlation between benzene and the other BTEX species, it's possible that this positive trend is due primarily to these underpredictions of high benzene levels coinciding with high levels of other BTEX compounds. In which case the estimates themselves may still be fairly robust to changing levels of VOCs. This is supported by the 75[th] percentile values, which do not change substantially as the levels of other VOCs vary from roughly 10 – 75 ppbC. However, n Figure 7d, the negative trends suggest that over-predictions by the model may be driven by cross-sensitivities that are not adequately corrected for as these over-predictions correspond to higher summed NMVOC concentrations. In this instance, higher levels of NMVOCs driving overestimations of methane highlight the need to improve the calibration model or utilize other approaches such as additional sensor signals to ensure that enhanced methane is truly methane and not a confounding compound. However, Figures 7a and 7d plots still do not display any clear patterns, such as a well-defined linear relationship, again indicating some robustness amid changing VOC compositions. The supplemental (Figures S4) contains plots of the Model 2 residuals versus individual VOC species. Comparing the residuals to individual species rather than the summed totals better reveals which confounding

species seem to be influencing our estimates. For example, positive linear trends in residuals present in both the benzene and summed aromatics Model 2 reveal that the behaviour of the aromatic species (benzene, toluene, $C_8$ alkylbenzenes, and $C_9$ alkylbenzenes) are not fully captured by our models. However, for the residuals of both the benzene and summed aromatics Model 2 estimates, individual OVOCs do not display any sort of trend or pattern in the residuals.

In an effort to further understand model robustness and specialization, we explored model fit statistics during times when the target VOCs were well versus poorly correlated with a potentially confounding VOC species. For this analysis, we applied a 1-hour moving window to the data and calculated (1) the $R^2$ between the target VOC(s) and another/other VOC(s), (2) the RMSE for our Model 2 results for that hour, and (3) the average target VOC(s) concentration for that hour. These plots, Figure 8, help to confirm whether or not our models are being specialized to our target VOC(s). For example, if our models are

predicting VOCs in a more general sense and not specializing for our targets, we would expect lower RMSE values to correspond to higher $R^2$ values and higher RMSE values to correspond to lower $R^2$ values. Lower $R^2$ values would correspond to periods of greater differentiation between VOC signals, potentially corresponding to higher RMSE values if the signals our models are predicting are more general. Conversely, if the models are being specialized to the target VOCs we would expect to see RMSE values that are more or less independent of correlations between different target VOC(s).

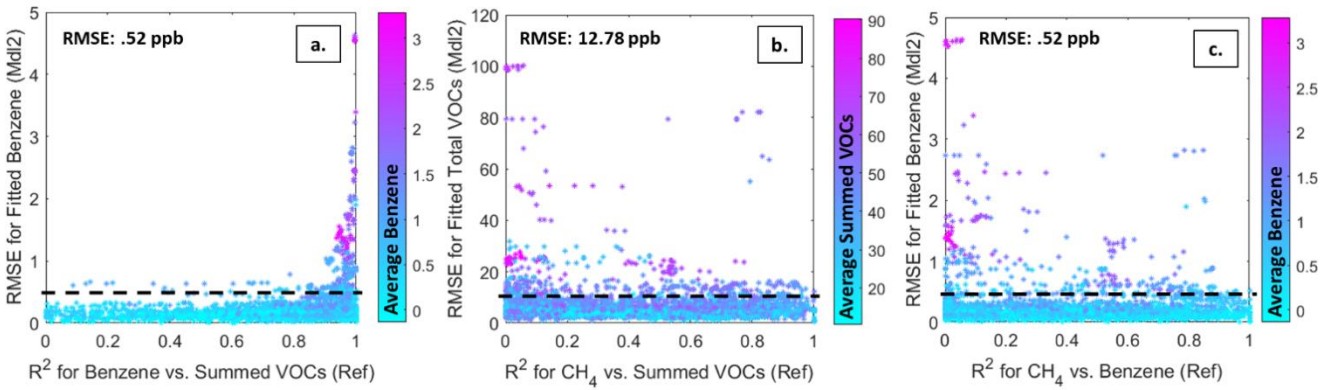

**Figure 8: Plots of error (RMSE) versus the coefficient of determination ($R^2$) for the target VOC or group versus a non-target, potentially confounding VOC or group. These values are calculated on the basis of a 1-hour moving window and the points are colored according to the average of the target pollutant or group for that hour. The dotted black line on each is the overall RMSE determined in Section 3.1.**

For our Model 2 for benzene, most of the RMSE values for each hour are below the overall RMSE for the model. For the points that are above the overall RMSE, many of these are for the high benzene concentrations – the underpredicted peaks. Furthermore, these higher benzene RSME values actually occur when we see high correlation between the summed VOCs and benzene, suggesting that the model is not simply fitting a more general VOC signal. Conversely, the high RMSE values do occur when there is low correlation between benzene and methane (Figure 8c), but there is still no clear trend of high RMSE

for low correlation and low RMSE values for high correlation. For the summed VOCs, there is also not a clear relationship between correlation and RMSE values.

These results provide supporting evidence that the models are becoming specialized to the intended target VOC or group of VOCs. Table 4 provides further support for this point by illustrating lower correlation between fitted datasets for Model 2 versus Model 1 and when there is lower correlation between the reference datasets. The pollutant pair in Table 4 worthy of a closer look is the benzene and summed aromatics. In this case these two reference data sets are very highly correlated, and it's not possible to confirm that the sensors are predicting benzene specifically, or a more general BTEX signature. The additional plots of residuals for the benzene Model 2, in the supplemental, do indicate trends with regards to the benzene data and the other aromatic species. Therefore, while it seems that the models can be trained to predict specific or different groups of VOCs, for highly correlated species or groups (that the sensors are selective for) it may be more difficult to make this distinction. Sensor users should be careful to not over-assign meaning to signals that are more likely to be indicative of VOC types or groups rather than specific species.

**Table 4: Coefficient of determination (R) between reference data pairs and fitted sensor data pairs.**

| Pollutant Pair | PT-RMS Data | Fitted Sensor Data (Model 1) | Fitted Sensor Data (Model 2) |
|---|---|---|---|
| Benzene vs. VOCs | 0.906 | 0.936 | 0.877 |
| Benzene vs. Methane | 0.464 | 0.901 | 0.826 |
| Benzene vs. Aromatics | 0.957 | 0.989 | 0.987 |
| Methane vs. VOCs | 0.531 | 0.902 | 0.864 |

**3.2 Sensor Selectivity & Consistency**

The availability of speciated VOC data also allowed us to compare the selectivity of each sensor. Given that the manufacturer lists different compounds as target gases for each sensor, we expected a difference in the selectivity between the two VOC sensors. Furthermore, it's likely that differences in selectivity aided the specialization of the calibration models in Section 3.1. We used analysis of variance to determine what the differences in selectivities might be and the results of this analysis are listed in Tables 5 and 6. The application of this technique is similar to previous studies where it was used to determine the effects of confounding species (Collier-Oxandale et al., 2018b; Eugster & Kling et al., 2012). These tables list the results of multiple runs in which different variables were included to determine their ability to explain the variance in the raw sensor signals. For instance, the first run includes only environmental parameters and time and for both sensors this set of predictors leaves the highest percentage of variability to residual error. Additional runs include all available reference signals (Run 2), or all available TNMHCs (Run 3). The final run in each case selects one VOC out of highly correlated groups (e.g., benzene for the aromatics and acetaldehyde for the aldehydes) and also results in the smallest portion of variance left to residual error, hence providing the strongest sets of explanatory variables. Comparing the two tables, the consistently important explanatory variables are highlighted in light grey. In Table 5, we see that acetaldehyde, benzene, methane and carbon monoxide are consistently important predictors for the Figaro 2600 signal, with methane and carbon monoxide as the most important. In Table 6, the percent of variance in the Figaro 2602 sensor explained by benzene becomes more dominant, while carbon monoxide and the aldehydes are no longer important predictors. The lack of sensitivity to carbon monoxide for the Figaro

2602 sensor could be especially valuable in identifying the effects of this cross-sensitivity and correcting for it. It is important to note here that we do not expect the Figaro 2602 to be sensitive to pure methane (based on laboratory tests and manufacturer information), so it is probable that this response is driven by other light alkanes co-emitted and correlated with methane. Overall, this analysis confirms a difference in selectivities, which supports the idea that these two sensors can together be leveraged in VOC and source identification. Furthermore, while this analysis has been limited to two Figaro VOC sensors, we would expect these selectivities to be consistent given the high correlation we have observed among Figaro sensors of the same type in past studies (Collier-Oxandale et al., 2018b).

**Table 5: Analysis of variance results for the Figaro 2600 sensor signal ($R_s/R_0$).**

| Run | Actd. | Acet. | Benz. | $C_8$ | $C_9$ | Form. | Meth. | Tol. | $CH_4$ | CO | $CO_2$ | $O_3$ | $NO_2$ | $H_2S$ | Temp | AH | Time | Err |
|-----|-------|-------|-------|-------|-------|-------|-------|------|--------|------|--------|-------|--------|--------|-------|-------|------|------|
| 1 | - | - | - | - | - | - | - | - | - | - | - | - | - | - | 42.65 | 4.14 | 5.35 | 47.86 |
| 2 | 2.23 | 0.06 | 3.23 | 0.04 | 0.01 | 0.37 | 0.27 | 0.67 | 6.42 | 10.16 | 0.08 | 0.23 | 0.08 | 0.26 | 8.18 | 19.71 | 6.84 | 41.16 |
| 3 | 3.45 | 0.43 | 5.30 | 0.07 | 0.65 | 1.55 | 0.12 | 0.91 | - | - | - | - | - | - | 31.20 | 10.00 | 5.16 | 41.14 |
| 4 | 3.48 | 1.09 | 3.80 | 0.00 | 0.21 | 1.58 | 0.32 | 0.78 | 9.18 | - | - | - | - | - | 19.34 | 13.26 | 9.11 | 37.84 |
| 5 | 2.58 | 0.42 | 2.75 | 0.02 | 0.00 | 0.61 | 0.11 | 0.60 | 10.42 | 8.85 | - | - | - | - | 16.41 | 15.62 | 7.61 | 33.99 |
| 6 | - | - | - | - | - | - | - | - | 8.27 | 16.86 | 0.00 | 0.00 | 3.35 | 0.02 | 3.66 | 21.13 | 5.60 | 41.11 |
| 7 | - | - | - | - | - | - | - | - | - | 18.45 | 4.81 | 0.00 | 4.37 | 0.21 | 3.81 | 19.65 | 3.22 | 45.47 |
| 8 | 6.63 | - | 5.42 | - | - | - | 0.28 | - | 9.36 | 9.77 | - | - | - | - | 17.06 | 14.04 | 6.60 | 30.83 |

Additional ANOVA results are available in the supplemental (Figures S5), including results of this analysis conducted on subsets of the data to test the robustness of our conclusions. The different subsets include the complete data, day vs. night data, and periods of elevated concentrations of specific compounds. Essentially these figures reinforce the conclusions drawn from the results above. Even across different subsets of data, the Figaro 2602 seems to be more responsive to aromatic species and the methane signal, while lacking sensitivity to carbon monoxide and the OVOCs. The Figaro 2600 is consistently responsive to methane, aromatic compounds, carbon monoxide, and to a lesser extent the aldehyde species. For the subsets where we see low percents of variance explained by predictors, this may be due to relatively lower concentrations of the pollutants revealed to be important (Figure S5 illustrates the main differences in the subsets of data). As the conclusions drawn from the ANOVA results seem to be consistent across different runs and subsets of data, this supports the likelihood of consistency in sensor selectivities.

**Table 6: Analysis of variance results for the Figaro 2602 sensor signal ($R_s/R_0$).**

| Run | Actd. | Acet. | Benz. | $C_8$ | $C_9$ | Form. | Meth. | Tol. | $CH_4$ | CO | $CO_2$ | $O_3$ | $NO_2$ | $H_2S$ | Temp. | AH | Time | Err. |
|-----|-------|-------|-------|-------|-------|-------|-------|------|--------|------|--------|-------|--------|--------|--------|------|------|------|
| 1 | - | - | - | - | - | - | - | - | - | - | - | - | - | - | 16.42 | 3.34 | 1.77 | 78.47 |
| 2 | 0.07 | 0.89 | 1.91 | 0.18 | 0.00 | 0.36 | 0.39 | 0.07 | 3.95 | 1.39 | 0.60 | 0.72 | 0.53 | 0.12 | 40.91 | 3.38 | 3.83 | 40.68 |
| 3 | 0.79 | 1.16 | 3.31 | 0.02 | 0.91 | 0.92 | 0.95 | 0.11 | - | - | - | - | - | - | 34.16 | 3.71 | 7.43 | 46.53 |
| 4 | 0.54 | 0.42 | 1.47 | 0.02 | 0.23 | 1.06 | 0.54 | 0.03 | 12.32 | - | - | - | - | - | 44.37 | 2.48 | 3.54 | 32.96 |
| 5 | 0.39 | 0.60 | 1.17 | 0.04 | 0.12 | 1.36 | 0.65 | 0.01 | 12.21 | 0.74 | - | - | - | - | 44.54 | 2.38 | 3.90 | 31.90 |

| | | | | | | | | | | | | | | | | | | |
|---|---|---|---|---|---|---|---|---|---|---|---|---|---|---|---|---|---|---|
| 6 | - | - | - | - | - | - | - | - | 7.05 | 4.13 | 1.38 | 0.80 | 0.08 | 0.75 | 34.89 | 1.51 | 1.51 | 47.90 |
| 7 | - | - | - | - | - | - | - | - | 0.00 | 4.90 | 10.94 | 0.57 | 0.00 | 1.83 | 29.57 | 1.28 | 2.67 | 48.24 |
| 8 | 0.83 | - | 10.63 | - | - | - | 0.77 | - | 11.04 | 0.34 | - | - | - | - | 42.70 | 2.04 | 2.65 | 29.00 |

In addition to insights into selectivity these results also reiterate the importance of the cross-sensitivities to environmental parameters like temperature and humidity. As indicated in Tables 5 and 6 temperature and/or humidity often explain a greater percentage of variance in the sensor signal than the pollutants of interest. While the residuals from the regression analysis in Section 3.1 indicate that the models seem to be adequately correcting for temperature and humidity effects (Figures S2), the effects of these parameters are complex. We know that temperature can not only impact the rate of reactions occurring at the sensor surface, but also the rate of desorption (Schutze et al., 2017; Sun et al., 2012). This behaviour means that temperature has the potential to impact the rates of response and recovery for the sensors as well as the magnitude of responses. Figures S7 provides an impression of these complexities. However, even though the pollutants explain a smaller percentage of the variance in sensor signal, performing the regression analysis from Section 3.1 with the sensor data excluded illustrates the value of the signals from the MOx sensors. Table 7 lists the results of the regression analysis for each Model 1 with all VOC sensor data excluded; this table also includes the original results in parentheses. Relying solely on environmental sensor data results in a higher RMSE and a much lower $R^2$, particularly for the testing data. Plots in the supplemental (Figure S8) further illustrate how including the sensor signal significantly improves our ability to predict short-term enhancements in pollutant levels and our ability to accurately track diurnal patterns.

**Table 7: Model 1 regression statistics, excluding MOx sensor data (original results, including MOx sensor signals)**

| | Training | | Testing | |
|---|---|---|---|---|
| | $R^2$ | RMSE | $R^2$ | RMSE |
| **Benzene (ppb)** | 0.22 *(0.68)* | 0.55 *(0.35)* | 0.08 *(0.58)* | 0.83 *(0.58)* |
| **Aromatics (ppbC)** | 0.24 *(0.63)* | 12.6 *(8.81)* | 0.09 *(0.56)* | 17.72 *(12.33)* |
| **Summed VOCs (ppbC)** | 0.26 *(0.68)* | 15.13 *(9.95)* | 0.07 *(0.59)* | 20.72 *(13.38)* |
| **Methane (ppm)** | 0.42 *(0.75)* | 0.32 *(0.21)* | 0.22 *(0.58)* | 0.35 *(0.24)* |

To better understand the consistency of the observed differences in sensor selectivities, we utilized bootstrapping for model training while also moving through different combinations of VOCs as predictors. Here each scenario was run 25 times with 15% of the dataset (in three-hour blocks) randomly selected for training and the remaining 85% of the data used for testing. Figure 9 shows the results for the testing data. The intended prediction data sets were calculated by first summing the species in ppbC or ppmC and then normalizing the resulting sum to better make relative comparisons. These results continue to support the conclusions drawn thus far. The Figaro 2602 (noted as Fig$_2$) is better at predicting BTEX compounds than the Figaro 2600 (noted as Fig$_1$), however, the opposite is true when predicting methane and carbon monoxide. The poorest performance results from the predictions of summed, oxygenated VOCs. The model including both sensors and an interaction term nearly always provides the best results, indicating the power in leveraging the difference in selectivities between the two sensor types.

On a final note, regarding the consistency of these conclusions, this bootstrap analysis as well as the original regression analysis was repeated for the second U-Pod (P2) co-located at this site (the results are available in the supplemental, Figures S9 and S10). In these results we see similar trends and behavior, but with poorer performance and greater variability. This poorer performance is likely due to the more fragmented nature of the data, as well as the possibly that the intermittent power failures

affected the sensor signal enough to decrease the performance. Overall, the similarities in results suggest consistency within sensor types.

An important issue related to sensor consistency is drift. Sensors signals may drift over time due to losses in sensitivity, the effects of ambient conditions, or even siting impacts (Masson et al., 2015; Miskell at al., 2017). As an example of magnitude, a study using the Figaro 2600 sensor over the course of 3 months in rural Alaska calibrated the sensors for methane at the start

of the study and observed a drift of 0.01 and 0.008 ppm per week across the two deployed sensors (Eugster & Kling, 2012). The length of our deployment did not allow for a comprehensive look at drift, however, we did look briefly at inter-sensor variability. To do this, we compared the correlation coefficients (R) for the two co-located sensors for a 24-hour period at the beginning and end of the deployment. The R values for the Figaro 2600 were 0.955 and 0.953 at the beginning and end respectively; suggesting that if the signals are drifting, this drift may be consistent across sensor type. Conversely, The R

values for the Figaro 2602 were 0.611 and 0.835 at the beginning and end respectively. While it is difficult to explain these two differing trends given the limitations in terms of number of sensors and the length of the deployment, this example highlights the importance of quantifying or correcting for sensor drift, as well as, understanding inter-sensor variability. The results are mixed in the literature as well, with some studies demonstrating high correlational and among sensors of the same type (Sadighi et al., 2017) and others demonstrating high variability (Castell et al., 2017). As more field work is conducted

with low-cost sensors, it would also be beneficial for researchers to consider whether and how drift is affected by the differing compounds a VOC sensor is exposed to.

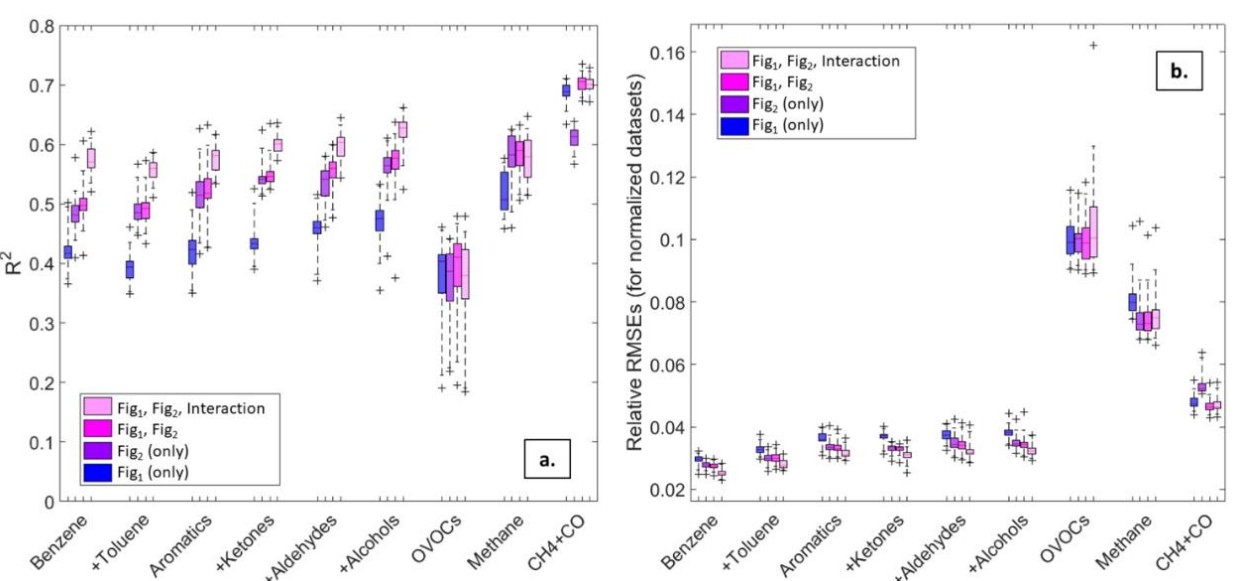

**Figure 9: The boxplots above illustrate the results of training on a randomly selected 15% of the data and testing on the remaining 85% 25 times, using each sensor individually, the two sensors as predictors, and the two sensors plus an interaction between them. Plots a and b depict the $R^2$ and RMSE respectively for the testing data. Note all of the data has been normalized and the whiskers are the 5th and 95th percentile respectively. The x-axis indicates the VOC or group being predicted.**

One final issue related to sensor consistency is understanding how robust sensor performance is to new environments. Given the constraints of this deployment we were not able to study sensor performance at different sites, but this will be an important area for future research. As was discussed in Section 2.1 not only do VOC levels vary across different environments, but also VOC compositions (Thompson et al., 2014). In new locations, both the environmental parameters and potentially confounding pollutants may vary as well. A few studies have considered the transferability of calibrations to new environments (Castell at

al., 2018; Malings et. al, 2018). One finding is that calibration models can overfit to the location where the model was trained (Vikram et al., 2019). Another study found that changes in the dominant local sources of pollutants may result in poorer performance for MOx ozone sensors (Casey & Hannigan, 2018). For this reason, it is important that training datasets encompass the conditions and pollutant levels that sensors will be exposed to at field sites.

### 3.3 Leveraging Sensor Differences to Learn About Potential Sources

Given the observed differences in selectivities, it is possible that sensor arrays may be able to provide useful information even in the absence of co-locations and field calibrations. For example, the ratios between different VOC sensors may be able to provide insight into VOC types or sources. This approach may be especially powerful if used in conjunction with methods such as passive sorption tubes; sensors could indicate emission events and a rough idea of VOC composition and then sampling tubes could provide more quantitative speciation.

Here we compared sensor ratios to reference pollutant ratios and trends. For this analysis, the baseline was identified and removed from both the sensor and the reference data using a technique applied to sensor data by Heimann and colleagues (2015). The purpose of this baseline removal was to isolate short-term emissions and remove the larger regional/diurnal trends. We then calculated the $R_s/R_0$ ratio as the Figaro 2600/Figaro 2602 and removed ratio values deemed 'insignificant'; insignificant values were identified as ratios where the data from one or both sensors was below a given threshold. The

threshold was 0.05 for the Figaro 2600 and 0.1 for the Figaro 2602. These thresholds were calculated as the average difference between paired values from the two co-located U-Pods P1 and P2. This was necessary as a low ratio could result from either the Figaro 2602 values being high or both the sensor values being low, in the latter case a low ratio may have been misleading. Finally, the different ratios were examined for correspondence to specific patterns in the reference data. Figure 10 below notes a few interesting ratios and trends. A complete look at these ratios in relation to reference data is available in the Figures S11.

While the results of this analysis are limited to this single deployment, they suggest that this approach has potential. In Figure 10a, we see that a low ratio of signals from Figaro 2600 (Fig$_1$) and Figaro 2602 (Fig$_2$), the Fig$_1$:Fig$_2$ ratio indicated in fuchsia, corresponds to a higher toluene to benzene ratio in the PTR-QMS data. As Halliday and colleagues (2016) observed, toluene to benzene ratios above 2.0 at this location are more likely to be the result of traffic, while lower ratios are more likely to be indicative of oil and gas emissions. If this VOC sensor ratio were to consistently identify periods with high toluene to benzene

ratios regardless of other concentrations and VOC compositions, this could be a powerful tool for differentiating between traffic and oil and gas emissions. Further supporting this point, a larger $Fig_1:Fig_2$ ratio indicated in yellow corresponds to data points that fall below the toluene to benzene ratio of 2.0, with a ratio of approximately 0.89. When examining this ratio, indicated in yellow, with regards to the benzene and methane reference data (Figure 10b), this $Fig_1:Fig_2$ ratio is similar to ratios

of benzene:methane observed in other studies occurring in oil and gas production areas (Helmig et al., 2014; Warneke et al., 2014; Petron et al., 2014). Thus, this analysis supports the idea that different sensor ratios may be indicative of emissions from different sources such as vehicles or oil and gas activity. Two final observations, in Figures 10a and 10c we see the $Fig_1:Fig_2$ ratio closest to a 1:1 relationship, indicated in green, both falls below the toluene:benzene ratio greater than or equal to 2.0 and corresponds to many of the enhancements in benzene that seem to occur independent of enhancements in carbon dioxide.

These observations suggest that this $Fig_1:Fig_2$ ratio, in green, is associated with relatively larger hydrocarbons (e.g., benzene) originating from volatilization or evaporation, rather than combustion. One possibility is that both the sensor ratio indicated in green and the sensor ratio indicated in yellow correspond to emissions from oil and gas activity, they may even relate to different activities or aspects of production. Though of course these observations would need to be demonstrated as consistent across different locations and with respect to differing background VOCs to ensure their reliability – the results here are limited

to this single time and place.

One point worth addressing is the poor correlation in the sensor ratios associated with the lower benzene:methane ratios (i.e., the points indicated in yellow in Figure 10b). However, applying an analysis technique developed by Halliday and colleagues (2018) provides additional support for the assertion that emissions from oil and gas activity are associated with this particular sensor ratio. To apply this technique, a rolling correlation was used to calculate (1) the correlation coefficients for each one

hour window, (2) the average benzene:methane ratio, (3) the average sensor ratio, and (4) the change in methane. Figure 11 depicts a histogram of the average sensor ratios for the complete dataset and for a subset in purple. The subset includes hourly averages with R values higher than 0.85, benzene:methane ratios < 1 ppb/ppm (the typical range observed in an oil and gas area), and changes in methane > 0.5 ppm. This technique allows us to essentially extract periods where emissions are likely the result of oil and gas activity (based on the benzene: methane ratio), are actively occurring (based on the correlation), and

are relatively large (based on the change in methane levels). As shown in Figure 11, the periods that meet these conditions seem to be associated with the sensor ratio indicated in yellow in Figure 10; further supporting the idea that sensor ratios may be able to assist with source identification.

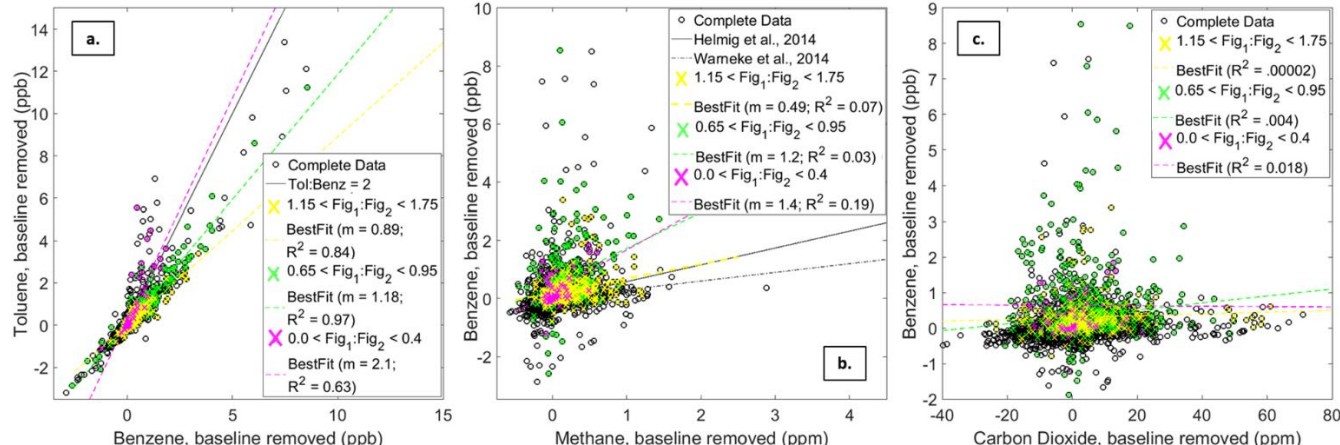

**Figure 10: Each panel displays two pollutants plotted together (from the reference data set). Panel a is the benzene versus the toluene while panel b is methane versus the benzene and panel c is carbon dioxide versus benzene. All of the points are plotted (black circles). Points corresponding to certain sensor ratios are then colored according to the colors listed in each the legend. Other relevant information, for example a ratio of 2 for the toluene to benzene ratio or relationships from previous studies.**

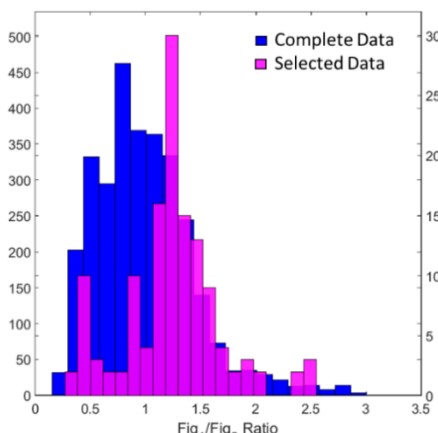

**Figure 11: Histogram of complete sensor ratios for each hour using moving correlation window vs. the ratios for a select subset of data with high correlation between benzene and methane (> 0.85), a ratio of benzene:methane typically associated with oil and gas activity (< 1.0 ppb/ppm), and a significant change in methane (> 0.5 ppm).**

## 4. Conclusion

While more field research is necessary to provide better insight into VOC sensor performance across new and different environments, here we have provided an overview of the potential for MOx VOC sensors. Not only were calibration models capable of providing concentration estimates relevant for ambient studies, but also these models appear to be specialized to the target pollutants and robust across changing compositions of other VOCs. Furthermore, this analysis confirmed a difference in selectivity between two MOx VOC sensors, a difference which can be leveraged in the development of calibration models,

to identify and mitigate cross-sensitivities, and potentially in source classification. Cross-sensitivities to confounding species are currently a major concern, for low-cost sensors in general and in particular for VOC sensing. However, given the differences in selectivity it seems that multiple sensors could be used to strategically determine the gases most likely affecting a sensor. For example, a carbon monoxide sensor and the Figaro 2602 could help to confirm whether methane is the main

driver of a response from the Figaro 2600. Furthermore, if the demonstrated association between sensor ratios and source types is shown to be consistent, multi-sensor devices could be a powerful tool for collecting preliminary or supplementary data in areas affected by numerous and complex sources – like environmental justice communities. These sensors can provide information at higher spatial and temporal resolution than is currently available. While there exist a number of uncertainties and limitations as discussed throughout this paper, through thoughtful use these tools may be able to begin to provide valuable

information as we also work to improve them. A few initial best practices to consider may include (1) calibrating sensors or characterizing their performance both before and after the field deployment in order to address drift, (2) ensuring that training data covers the conditions expected in the field in terms of both environmental conditions and the pollutant concentrations/compositions, and (3) learning as much as possible about potential cross-sensitivities and then incorporating multiple sensors of different selectivities into your platform.

Given their cost and the relative ease of deployment, these tools have the potential to provide information to support public health research, community-based environmental justice studies, or even supplement research by regulatory or academic communities. For example, by guiding exposure studies or providing a better idea of the impact of nearby sources on overburdened communities. Often in environmental justice communities, lacking resources, even cursory information on VOCs and local emissions could be valuable. These types of sensors could also supplement conventional monitoring

approaches. For example, regulatory agencies sometimes utilize TNMHC measurements and MOx sensors may be able to supplement these instruments again by providing greater spatial resolution. Multi-sensor systems could also provide time-resolved information, adding to data collected using a speciated method such as VOC canisters or passive sorption tubes. Future research will hopefully explore these applications as well as further quantify the capacity and limitations of these sensors, however, the usefulness demonstrated here speaks to the potential MOx sensors have to provide new insights into the

complex and dynamic VOC types and sources impacting our lives and communities.

**Data Code & Availability**

All sensor data (including final datasets and raw sensor data) and MatLab code used to process the data is available through the main author, please contact for access. All reference data was provided courtesy of the NATIVE Trailer team (Penn State University) and is available in the official DISCOVER-AQ database: https://www-air.larc.nasa.gov/missions/discover-

aq/discover-aq.html.

## Acknowledgments

Funding provided through the MetaSense Project (NSF Grant CNS-1446912), the AirWaterGas Project (NSF-SRN CBET: 1240584), and the DISCOVER-AQ Project (NASA). Thank you to all project partners during the DISCOVER-AQ/FRAPPE campaigns (NCAR, NOAA, CDPHE, US EPA). Additional thanks to research partners from the NATIVE trailer team (Penn State) and the Wisthaler group (University of Oslo). Special thanks also to all current and former members of the Hannigan Research Lab.

## Author Contributions

ACO led the low-cost sensor deployment and field work. HH operated and maintained the reference instruments at the field site. ACO, MH, and JT all contributed to work quantifying low-cost sensors for VOCs. ACO processed this dataset and led the analysis with feedback from all co-authors. ACO organized this manuscript with assistance on writing as well as feedback edits from all co-authors.

## Competing Interests

The authors declare that they have no conflicts of interest.

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
