# Peer review of "Understanding the ability of low-cost MOx sensors to quantify ambient VOCs"

_Atmospheric Measurement Techniques, 2018_

## Referee Comment (RC1) · Anonymous Referee #1 · 29 Oct 2018

Review of Collier-Oxendale et al. "Understanding the ability of low-cost MOx sensors to quantify ambient VOCs"

This paper is a welcome addition to the literature on the issue of understanding and utilising low-cost sensors systems for measuring and monitoring trace gas concentrations in the atmosphere. The focus of the paper is on the co-location of the systems with more accepted independent measurement techniques for the molecules of interest, i.e. VOCs and selected inorganics. The authors describe the statistical methods and techniques in detail and show comprehensive details of the analysis of the uncertainties and effective of their methods in this deployment.

More detailed required on the co-located methods.

[Figure]

The whole analysis falls or stands of the methods used to compare the sensor data. It is therefore important that the correct details, with references, of the instrumentation used in this study. For example, what type of ptr-ms? ToF/quad? Was the LGR instrument really a cavity ring-down or was it an off-access output spectrometer. This may seem picky, but this is techniques and instrumentation paper and so the details need to be spelled out.

Discussion on applicability of techniques described and warnings regarding using these type of sensors in isolation with training data in new environments.

The study data and methods are well analysed and described. However the sensors are trained using very similar data to the data generated by the methods described and the sensors used. How do we know the accuracy of the sensors in new environments? Techniques used in atmospheric analysis are commonly found to have interferences in "new" environments that result in many years of inferences on "false" data. Low-cost sensors is a new area and it would be good for well-run studies such as this to provide some words of warning for the deployments outside of co-location of well-characterised methods, in a spatial mesh to provide high resolution spatial data to a fixed high quality measurement location or well-understood environments. It is important to note that models only work within the parameters of the training data and so phenomena that are encountered outside of the scope of the training data will probably have unknown uncertainties associated. More in depth discussion of these issues would be of benefit in addition to few lines in the conclusions.

Would authors consider adding a "Best practices and procedures" section to the discussion section as mentioned on page 5 section 1.1? This would be a usable summary of their findings.

Comments, data or discussion of the lifetime of MOx sensors and the sensitivity drift.

Do MOx sensors drift? How much do they drift? Is the drift similar across all detected molecules? The training data bookends the measured data and so any sensitivity

or selectivity drifts occurring will be implicit in the models produced by the training methods. In my experience there is significant drift and individual sensor-to-sensor variability even within batches. The authors should address the issue of sensor drift somehow within the limits of their study but also more generally how sensor drift issues may impact the application of low-cost sensor systems.

---

## Referee Comment (RC2) · Anonymous Referee #2 · 6 Nov 2018

This manuscript presents a field evaluation of two different metal oxide (MOx) sensors for volatile organic compound (VOC) detection, comparing the sensor signals with speciated VOC measurements from a proton-transfer mass spectrometer (PTR-MS). Measurements were made for approximately 3 weeks as part of a larger air quality study at the Platteville Atmospheric Observatory, Colorado, which is in close proximity to extensive oil and gas activity. The analysis is detailed and well written, and the work is a good addition to the available literature on these devices.

My main comment on the work is how applicable the conclusions are to more typical environments. The proximity to large emissions of VOCs from oil and gas activity makes for a very favorable environment for the MOx sensors being evaluated, due to the elevated VOC mixing ratios and the large dynamic range observed. The authors

suggest that these technologies could be used more widely for monitoring public health exposure, however, even in polluted urban environments mixing ratios of VOCs can be at least an order of magnitude lower than observed in this study. For example, Warneke et al. (2013) show benzene data from Los Angeles, from both ground and airborne measurement platforms, where all measurements are below the suggested 0.5 ppb lower limit for data exclusion proposed by the authors in Section 3.1.4. It would be very useful for the reader if the authors commented more on this and repeated some of the comparisons with the PTR-MS for only data in a range that would be comparable with a more typical urban environment.

Minor comments:

Page 9 lines 15:23: What was the motivation behind choosing ppbC instead of ppbV for the summed compound comparisons? In theory which unit gives better agreement should depend on the mechanism of the sensor interaction with the compounds. Unless the sensor converts a fixed fraction of a VOC to $CO_2$ and $H_2O$ does the use of ppbC not weight the signal from the larger compounds more than the smaller ones? The authors should comment on this, as if the use of ppbV instead of ppbC changes the agreement shown in Figures 2-5 it would suggest something about the sensor sensitivities to different VOCs in the groupings.

Table 3: More details required on why the models used were chosen. Was the formulation of these models informed by experiments or are they the best performing from a larger selection of randomly selected models?

Figures 2-5: Although the bootstrapping shown in Fig. 9 illustrates well the sensitivity of the model performance to the choice of training and test data, a statement in Sect. 3.1 on how sensitive model performance is to the choice of training data would be informative.

Page 11 line 11: Proton-transfer within the PTR-MS is also a chemical reaction. The slower time response could be due to a slower surface reaction on the MOx sensor but

more likely due to the diffusion control of gas sampled to the sensor surface.

Figure 7: The authors should explain the reason for the choice of values held constant (0.75 and 0.05 ppb for benzene, 19 and 38 ppb for summed aromatics, and 2.5 and 3 ppm for methane). The authors need to explain the reasoning behind choosing these values and the effect it has on the conclusions of this section of the paper.

SI: SI plots require more descriptive captions.

References Warneke, C., et al. (2013), Photochemical aging of volatile organic compounds in the Los Angeles basin: Weekday-weekend effect, J. Geophys. Res. Atmos., 118, 5018–5028, doi:10.1002/jgrd.50423.

---

## Author Response (AR1)

**Responses to Comments and Revised Manuscript:**
*Understanding the ability of low-cost MOx sensors to quantify ambient VOCs (Collier-Oxandale et al.)*

**Author Response to Reviewer 1 Comments:**

COMMENT: "Review of Collier-Oxandale et al. "Understanding the ability of low-cost MOx sensors to quantify ambient
5  VOCs" This paper is a welcome addition to the literature on the issue of understanding and utilizing low-cost sensors systems
for measuring and monitoring trace gas concentrations in the atmosphere. The focus of the paper is on the co-location of the
systems with more accepted independent measurement techniques for the molecules of interest, i.e. VOCs and selected
inorganics. The authors describe the statistical methods and techniques in detail and show comprehensive details of the analysis
of the uncertainties and effective of their methods in this deployment. More detailed required on the co-located methods."

RESPONSE: The authors appreciate the overview provided by the reviewer and would like to thank the reviewer for their
productive comments, which helped to improve this paper.

COMMENT: "The whole analysis falls or stands of the methods used to compare the sensor data. It is therefore important
15  that the correct details, with references, of the instrumentation used in this study. For example, what type of ptr-ms? ToF/quad?
Was the LGR instrument really a cavity ring-down or was it an off-access output spectrometer. This may seem picky, but this
is techniques and instrumentation paper and so the details need to be spelled out."

RESPONSE: The authors agree regarding the importance of precisely describing the reference instrumentation relied upon in
20  this analysis. To address this comment, additional details and expanded descriptions have been added to Section 2.2.

COMMENT: "Discussion on applicability of techniques described and warnings regarding using these type of sensors in
isolation with training data in new environments. The study data and methods are well analysed and described. However the
sensors are trained using very similar data to the data generated by the methods described and the sensors used. How do we
25  know the accuracy of the sensors in new environments? Techniques used in atmospheric analysis are commonly found to have
interferences in "new" environments that result in many years of inferences on "false" data. Low-cost sensors is a new area
and it would be good for well-run studies such as this to provide some words of warning for the deployments outside of co-
location of well-characterised methods, in a spatial mesh to provide high resolution spatial data to a fixed high quality
measurement location or well-understood environments. It is important to note that models only work within the parameters
30  of the training data and so phenomena that are encountered outside of the scope of the training data will probably have unknown

uncertainties associated. More in depth discussion of these issues would be of benefit in addition to few lines in the conclusions."

RESPONSE: This is a significant point raised by the reviewer and the authors agree that it is important to highlight the similarity of the training and testing datasets used in this study as well as discuss potential limitations. To further bring attention to this issue the authors have added a discussion to Section 3.2 on the issue of the transferability of calibrations for low-cost sensors to new environments including referencing existing literature on the topic. The authors also note the potential complexity of this issue for VOC sensors, as in addition to variation in the environmental parameter space and pollutant ranges, the composition of VOCs is likely to vary in new environments.

COMMENT: "Would authors consider adding a "Best practices and procedures" section to the discussion section as mentioned on page 5 section 1.1? This would be a usable summary of their findings."

RESPONSE: The authors agree that best practices and procedures are especially important for low-cost sensors as they are such as accessible technology. While this study is more of a preliminary look at using VOC sensors in the field, several overarching, guiding best practices have been added to the conclusion section. These have been framed as initial best-practices and they provide a foundation for future studies to expand upon.

COMMENT: "Comments, data or discussion of the lifetime of MOx sensors and the sensitivity drift. Do MOx sensors drift? How much do they drift? Is the drift similar across all detected molecules? The training data bookends the measured data and so any sensitivity or selectivity drifts occurring will be implicit in the models produced by the training methods. In my experience there is significant drift and individual sensor-to-sensor variability even within batches. The authors should address the issue of sensor drift somehow within the limits of their study but also more generally how sensor drift issues may impact the application of low-cost sensor systems."

RESPONSE: The authors certainly agree regarding the importance of characterizing and understanding drift for low-cost sensors. In an effort to highlight this issue, a discussion has been added to Section 3.2 that explores why drift occurs and what other researchers have observed using low-cost sensors. Given the length of the field deployment for this dataset, we were not able to characterize long-term drift. However, we did examine inter-sensor variability at the start and end of the measurement campaign in an effort to understand if any drift might be consistent across sensors of the same type. The results were mixed; we found high and consistent correlation among the two Figaro 2600 sensors at the start and end of the campaign, however, we found low correlation among the two Figaro 2602 sensors at the start and higher correlation at the end. Given the limitations of the dataset, it's difficult to explain what drove the difference in these results, but we believe that including these results illustrates the importance of considering inter-sensor variability, drift, and losses in sensitivity during sensor studies. As to the

reviewer's point regarding whether VOC sensors drift consistently across different compounds – this is a very interesting question and the authors have indicated that this would be valuable future work.

**Author Response to Reviewer Comment 2:**

COMMENT: "This manuscript presents a field evaluation of two different metal oxide (MOx) sensors for volatile organic compound (VOC) detection, comparing the sensor signals with speciated VOC measurements from a proton-transfer mass spectrometer (PTR-MS). Measurements were made for approximately 3 weeks as part of a larger air quality study at the Platteville Atmospheric Observatory, Colorado, which is in close proximity to extensive oil and gas activity. The analysis is detailed and well written, and the work is a good addition to the available literature on these devices."

RESPONSE: The authors appreciate the overview provided by the reviewer and would like to thank the reviewer for their insightful comments, which helped to strengthen this

COMMENT: "My main comment on the work is how applicable the conclusions are to more typical environments. The proximity to large emissions of VOCs from oil and gas activity makes for a very favorable environment for the MOx sensors being evaluated, due to the elevated VOC mixing ratios and the large dynamic range observed. The authors suggest that these technologies could be used more widely for monitoring public health exposure, however, even in polluted urban environments mixing ratios of VOCs can be at least an order of magnitude lower than observed in this study. For example Warneke et al. (2013) show benzene data from Los Angeles, from both ground and airborne measurement platforms, where all measurements are below the suggested 0.5 ppb lower limit for data exclusion proposed by the authors in Section 3.1.4. It would be very useful for the reader if the authors commented more on this and repeated some of the comparisons with the PTR-MS for only data in a range that would be comparable with a more typical urban environment."

RESPONSE: The authors agree with the reviewer that this is an important point. To address this comment, a discussion of the potential for elevated levels of VOCs in oil and gas production areas and how this may compare to typical urban areas has been added to Section 2.1. Additional discussion of sensor performance considering the levels of VOCs observed in this study as compared to what might be seen in other environments has also been added to Section 3.1.4. As per the reviewer's suggestion, the authors also ran the regression analysis for Models 1 and 2 for benzene, utilizing only values under 0.75 ppbV. The resulting figures are available in the Supplemental Materials (Supplemental Figure S3) and a comment regarding this additional analysis has been added to the text. Essentially, correlation to the reference data remains high – suggesting that it may be possible to train models for lower detection limits. However, it's also possible that the benzene signal is being enhanced by the other well-correlated aromatics that the sensors seem well-suited to detect. Further study will be necessary to confirm whether lower detection limits are possible to achieve in field settings.

COMMENT: "Page 9 lines 15:23: What was the motivation behind choosing ppbC instead of ppbV for the summed compound comparisons? In theory which unit gives better agreement should depend on the mechanism of the sensor interaction with the compounds. Unless the sensor converts a fixed fraction of a VOC to $CO_2$ and $H_2O$ does the use of ppbC not weight the signal from the larger compounds more than the smaller ones? The authors should comment on this, as if the use of ppbV instead of ppbC changes the agreement shown in Figures 2-5 it would suggest something about the sensor sensitivities to different VOCs in the groupings."

RESPONSE: This is an important observation and was a point of discussion among the authors as well. The authors chose to use ppbC, as opposed to ppbV, as it seemed like the more appropriate way to sum the VOC signals because it does take in to ac- count the differences in individual VOC compound sizes. Given the sensor operating principles (i.e., a chemical reaction on the sensor surface), we would expect the extent of the sensor response to vary based on the size as well as the make-up of the VOC compound. When we fit to a summed ppbC signal, we are fitting to a signal that incorporates some of these important differences. Furthermore, the authors did begin by summing the ppbV values for each VOC species and running the same analysis. The result was no change for the BTEX summed signal – likely due to the fact that these signals were all highly correlated, so the magnitude of the signal changed but not the overall trends. There was, however, a slight improvement for Model 1 for the summed VOC signal when the ppbC values were used as opposed to the ppbV values. This improvement was likely due to the ppbC sum being weighted for the larger BTEX compounds. This signal placed a smaller emphasis on the oxygenated VOC compounds, which as suggested by the results presented in Figure 9, the sensor may be less well-suited to predict. The authors have added further rationale for this choice to Section 2.4 in the hopes of clarifying this decision for readers.

COMMENT: "Table 3: More details required on why the models used were chosen. Was the formulation of these models informed by experiments or are they the best performing from a larger selection of randomly selected models?"

RESPONSE: A brief explanation has been added to Section 2.4 in an attempt to clarify why these models were selected. The reasoning was to begin with models typically used to calibrate low-cost air quality sensors, as this is a more preliminary look at VOC sensors in the field. Additional predictors were added to attempt to address issues with the models (e.g., patterns in the residuals). These added predictors were found through trial and error.

COMMENT: "Figures 2-5: Although the bootstrapping shown in Fig. 9 illustrates well the sensitivity of the model performance to the choice of training and test data, a statement in Sect. 3.1 on how sensitive model performance is to the choice of training data would be informative."

RESPONSE: When this same analysis was run displaying the model performance results for the training data rather than that testing data, it produced similar trends. The main difference was improvements to the performance statistics (i.e., a higher R-squared and a lower RMSE values). While the authors agree that model performance with respect to training data is an important consideration, we are not sure this study provides the best dataset to examine this issue, given the relatively short

5 length of the deployment (e.g., it does not span any seasonal changes) and that there are no periods where the site was obviously subject to a distinctly different mixture of VOCs, thus we have not added any figures at this point. That being said, the authors would be happy to add the results of this analysis with respect to different selections of training data to either the main paper or the appendix if it would be valuable to readers.

10 COMMENT: "Page 11 line 11: Proton-transfer within the PTR-MS is also a chemical reaction. The slower time response could be due to a slower surface reaction on the MOx sensor but more likely due to the diffusion control of gas sampled to the sensor surface."

RESPONSE: The authors agree that diffusion is an important factor in the sensor's exposure and response to the target
15 pollutants and the statement has been reworded to highlight it's role.

COMMENT: "Figure 7: The authors should explain the reason for the choice of values held constant (0.75 and 0.05 ppb for benzene, 19 and 38 ppb for summed aromatics, and 2.5 and 3 ppm for methane). The authors need to explain the reasoning behind choosing these values and the effect it has on the conclusions of this section of the paper."

RESPONSE: In order to address this comment an explanation as to why the 75th and 95th percentile values were chosen has been added to the text prior to Figure 7. Then, following Figure 7, the discussion of the results depicted was expanded to examine the implications of these results for sensor use. In short, the reason for this selection was based on the assumption that a well-suited use of low-cost VOC sensors, at least for now, is for the detection of relatively large enhancements above
25 background or typical levels. The results of the analysis depicted in Figure 7 have implications for use of sensors for in particular applications.

COMMENT: "SI: SI plots require more descriptive captions."

30 RESPONSE: The authors have expanded the descriptions in the captions for all the plots in the Supplemental, in an effort the clarify these additional materials.

References Warneke, C., et al. (2013), Photochemical aging of volatile organic compounds in the Los Angeles basin: Weekday-weekend effect, J. Geophys. Res. Atmos., 118, 5018–5028, doi:10.1002/jgrd.50423.

*Revised Manuscript Follows with Revisions Indicated in 'Track Changes' and Noted with Comments:*

**Understanding the ability of low-cost MOx sensors to quantify ambient VOCs**

5   Ashley M. Collier-Oxandale[1], Jacob Thorson[2], Hannah Halliday[3], Jana Milford[2], and Michael Hannigan[2]

[1]Environmental Engineering, University of Colorado Boulder, Boulder, 80309, USA
[2]Mechanincal Engineering, University of Colorado Boulder, Boulder, 80309, USA
[3]NASA Langley Research Center, Hampton, VA, 23666, USA

10   *Correspondence to*: Ashley M. Collier-Oxandale (ashley.collier@colorado.edu)

**Abstract.**

Volatile organic compounds (VOCs) present a unique challenge in air quality research given their importance to human and environmental health, and their complexity to monitor resulting from the number of possible sources and mixtures. New technologies, such as low-cost air quality sensors have the potential to support existing air quality measurement methods by
15   providing high time and spatial resolution data. This higher resolution data could provide greater insight into specific events, sources, and local variability. Furthermore, given the potential for differences in selectivities for sensors, leveraging multiple sensors in an array format may even be able to provide insight into which VOCs or types of VOCs are present. During the FRAPPE/DISCOVER-AQ monitoring campaigns, our team was able to co-locate two sensor systems, using metal oxide (MOx) VOC sensors, with a proton-transfer-reaction mass spectrometer (PTR-MSPTR-QMS) providing speciated VOC data.
20   This dataset provided the opportunity to explore the ability of sensors to estimate specific VOCs and groups of VOCs in real-world conditions, e.g., dynamic temperature and humidity. Moreover, we were able to explore the impact of changing VOC compositions on sensor performance as well as the difference in selectivities of sensors in order to consider how this could be utilized. From this analysis, it seems that systems using multiple VOC sensors are able to provide VOC estimates at ambient levels for specific VOCs or groups of VOCs, it also seems that this performance is fairly robust to changing VOC mixtures,
25   and it was confirmed that there are consistent and useful differences in selectivities between the two MOx sensors studied. While this study was fairly limited in scope, the results suggest that there is the potential for low-cost VOC sensors to support highly resolved, ambient hydrocarbon measurements. The availability of this technology could enhance research and monitoring for public health and communities impacted by air toxics, which in turn could support a better understanding of exposure and actions to reduce harmful exposure.

> **Commented [a1]:** PTR-MS has been changed to PTR-QMS throughout to add specificity to the instrument used

[revised manuscript text omitted]

**Commented [a2]:** Here a discussion on the potential for enhanced VOCs in oil and gas production areas has been added. (Reviewer 2)

[revised manuscript text omitted]

**Commented [a4]:** A brief explanation of our choice to use p for the summed signals has been added here, as well as a acknowledgment of how this might affect the results (Reviewer 2

predictors were added to improve the resulting statistics and residuals for a given target pollutant or pollutant group. In addition to simulating a field calibration, we also utilized bootstrapping and analysis of variance to get a more fundamental understanding of the sensors' selectivities and the consistency of their behavior. Model 1 was selected for this analysis as it is similar to multiple linear regression models typically used when exploring sensor performance (Spinelle et al., 2015; Zimmerman et al., 2018; Casey et al., 2019). The additional predictors added to Model 2, determined through trial and error, facilitate a look into whether there might be potential to improve these models by addressing the patterns in the residuals.

**Table 3: Multiple linear regression models used**

| Model Identifier | Model |
|---|---|
| Model 1: for all | $C = p_1 + p_2 * VOC_1 + p_3 * VOC_2 + p_4 * (VOC_1 * VOC_2) + p_5 * Temp. + p_6 * Abs.Hum. + p_7 * Time$ |
| Model 2: for benzene | $C = p_1 + p_2 * VOC_1 + p_3 * VOC_2 + p_4 * (VOC_1 * VOC_2) + p_5 * Temp. + p_6 * Abs.Hum. + p_7 * Time$ $+ p_8 * VOC_2(V) + p_9 * (Temp.* VOC_2)$ |
| Model 2: for aromatics | $C = p_1 + p_2 * VOC_1 + p_3 * VOC_2 + p_4 * (VOC_1 * VOC_2) + p_5 * Temp. + p_6 * Abs.Hum. + p_7 * Time$ $+ p_8 * VOC_2(V) + p_9 * (Temp.* VOC_2)$ |
| Model 2: for VOCs | $C = p_1 + p_2 * VOC_1 + p_3 * VOC_2 + p_4 * (VOC_1 * VOC_2) + p_5 * Temp. + p_6 * Abs.Hum. + p_7 * Time$ $+ p_8 * (\ln(Temp.) * Abs.Hum.)$ |
| Model 2: for methane | $C = p_1 + p_2 * VOC_1 + p_3 * VOC_2 + p_4 * (VOC_1 * VOC_2) + p_5 * Temp. + p_6 * Abs.Hum. + p_7 * Time$ $+ p_8 * CO2$ |

**Model predictors: VOC$_1$ – Figaro 2600 R/R0, VOC$_2$ – Figaro 2602 R/R$_0$, Temp – temperature (degrees C), Abs. Hum. – absolute humidity, Time – continuous time, VOC$_2$(V) – Figaro 2602 voltage signal, CO$_2$ – carbon dioxide concentration (in this case from the reference data); C is the concentration value being solved for, either an individual or group of VOCs**

Clarifying how target VOCs and groups of VOCs were selected: benzene and summed aromatic species (including C$_8$ and C$_9$ alkylbenzenes) were selected for health reasons, as discussed in Section 1. While benzene health risks are the most well understood, the other common aromatic species (e.g., the BTEX compounds: benzene, ethylbenzene, toluene, and xylene) also present similar concerns for human health (Adgate et al., 2014). The summed total VOC signal was selected to provide some insight into the sensors' capacity to predict total non-methane organic compounds (TNMOCs) or possibly total non-methane hydrocarbons (TNMHCs). This signal was calculated by summing the ppbC values for all of the available species measured by the PTR-QMS: acetaldehyde, acetone, benzene, C$_8$ alkylbenzenes, C$_9$ alkylbenzenes, formaldehyde, methanol, and toluene. This type of measurement may be useful in an area concerned with a broad array of air toxics or when used in combination with a method of VOC speciation.

**Commented [a5]:** A brief explanation has been added here to clarify why these models were selected. (Reviewer 2)

**Commented [a6]:**

**3. Results & Discussion**

**3.1 Field Calibration Performance**

In the following sections, we show the results of the two MLR models for predicting each of the target VOCs or groups of VOCs. In each case, a timeseries is included to illustrate qualitatively the models' ability to predict trends and VOC concentrations, while regression statistics note the performance of the model across training and testing data and any changes from Model 1 to Model 2. The training periods have been highlighted in yellow and are the same training and testing periods used in the previous methane quantification work (Collier-Oxandale, 2018b). Also included are scatterplots to highlight improvements in testing data from Model 1 to Model 2, and boxplots of the residuals (observed – predicted) to show where the majority fall (despite the wide variance apparent in the scatterplot). Additional residual plots are available in the supplemental (Figures S2)

**3.1.1 Estimating Benzene and Summed Aromatic Species**

Figures 2 and 3 present the results for benzene and the summed aromatic level quantification. Overall, both models capture the diurnal trends and short term elevated concentrations, although the models under predict the highest concentration events. Model 2 performs better than Model 1, with $R^2$ values of 0.67 and 0.64 for benzene and summed aromatics respectively. In both cases, Model 2 pulls some of the more extreme values closer to the 1:1 line. For the aromatics, Model 2 also results in closer fitting values at low concentrations. Furthermore, the RMSE values for the Model 2 testing data, 0.52 ppb and 11.25 ppbC for benzene and summed aromatics respectively, are less than the dynamic range observed in this dataset suggesting estimations can be made at the ambient levels observed during this deployment. The underprediction of the benzene and summed aromatic peaks is most likely due to a limitation associated with sensor response time. The response time for the PTR-QMS has a 1 second per species integration time during the 1-minute measurement cycle (Gouw & Warneke, 2007), however the MOx sensors respond more slowly as they are relying  not only on a chemical reaction on the surface of the sensor, but also the diffusion of the target gas to that surface. This means that a sensor may not be able to reach steady state in the time it takes for a plume to pass.

> **Commented [a7]:** This statement has been reworded to incorporate reviewer 2's comment of the role of diffusion in the sensor's exposure to the target gas. (Reviewer 2)

[revised manuscript text omitted]

**Commented [a11]:** Here we have added a discussion of sensor drift. We have clarified what we can and cannot learn from this dataset as well as highlighted issues to be explored in future research into low-cost VOC sensors specifically. (Reviewer 1)

[Figure]

**Figure 9: The boxplots above illustrate the results of training on a randomly selected 15% of the data and testing on the remaining 85% 25 times, using each sensor individually, the two sensors as predictors, and the two sensors plus an interaction between them. Plots a and b depict the $R^2$ and RMSE respectively for the testing data. Note all of the data has been normalized and the whiskers are the 5th and 95th percentile respectively. The x-axis indicates the VOC or group being predicted.**

One final issue related to sensor consistency is understanding how robust sensor performance is to new environments. Given the constraints of this deployment we were not able to study sensor performance at different sites, but this will be an important area for future research. As was discussed in Section 2.1 not only do VOC levels vary across different environments, but also VOC compositions (Thompson et al., 2014). In new locations, both the environmental parameters and potentially confounding pollutants may vary as well. A few studies have considered the transferability of calibrations to new environments (Castell at al., 2018; Malings et. al, 2018). One finding is that calibration models can overfit to the location where the model was trained (Vikram et al., 2019). Another study found that changes in the dominant local sources of pollutants may result in poorer performance for MOx ozone sensors (Casey & Hannigan, 2018). For this reason, it is important that training datasets encompass the conditions and pollutant levels that sensors will be exposed to at field sites.

**3.3 Leveraging Sensor Differences to Learn About Potential Sources**

Given the observed differences in selectivities, it is possible that sensor arrays may be able to provide useful information even in the absence of co-locations and field calibrations. For example, the ratios between different VOC sensors may be able to provide insight into VOC types or sources. This approach may be especially powerful if used in conjunction with methods such as passive sorption tubes; sensors could indicate emission events and a rough idea of VOC composition and then sampling tubes could provide more quantitative speciation.

Here we compared sensor ratios to reference pollutant ratios and trends. For this analysis, the baseline was identified and removed from both the sensor and the reference data using a technique applied to sensor data by Heimann and colleagues

**Commented [a12]:** Here a discussion of considering how well sensor performance holds up in new environments has been added. (Reviewer 1)

[revised manuscript text omitted]

**Commented [a13]:** New text was added to the Conclusion section to reiterate the important observations from both reviewers regarding the transfer of sensors to new environments and the use of sensors in environments with appropriate VOC ranges. Some initial best practices were also added, as suggested by Reviewer 1, the authors thought they should be framed as initial, given that this is an early field study of VOC sensors, and there is a lot left to learn about their capabilities and suitable applications. (Reviewers 1 and 2)

[revised manuscript text omitted]

Commented [a15]: Detail has been added to all captions

**Supplemental Materials**

***Figure S1 – Correlation** plots for the minute-resolution data from all the NATIVE Trailer instruments*

[Figure]

***Figures S2 -  Regression model residuals (from Section 3.1)**; the following plots depict the residuals from the Model 1 and Model 2's for each benzene (a: top left four panels), summed aromatics (top right four panels), summed VOCs (bottom left four panels),* and methane (d: bottom right four panels); the residuals for the training data for each model are plotted vs. the pollutant concentration being estimated, temperature, humidity values, and time.*

5

[Figure]

a.

[Figure]

b.

[Figure]

c.                                                              d.

*Figure S3 – Linear rRegression aAnalysis for bBenzene < 0.75 ppbV; these results are for the same analysis as was conducted in Section 3.1 .1 using benzene as the intended predictor and excluding all benzene concnetrations over 0.75 ppbV; note the whiskers indicted in red in the boxplots (panel c) represent the 95th percentile values for the residuals*

**Commented [a16]:**

[Figure]

*Figures S4 – Additional residuals for models vs. target and non-target VOCs (Section 3.1.4); here the residuals are plotted against gases that are not the intended predictor in an effort to understand whether the estimates predicted by the calibration models are robust tot changing levels of other VOCs and potential confounders, again roughly the 75th and 95th percentile*

[Figure]

[Figure]

*Figures S5 – ANOVA results illustrating the percentage of variance in the sensor signal explained by various predictors for complete data and subsets of data (top – Figaro 2600, bottom – Figaro 2602) (Section 3.2); the colors indicate the percent of variability explained by a particular predictor (listed along the x-axis); the subsets of data (defined in Section 3.2) are the complete data, day-time data, night-time data, and selected periods with relatively higher amounts of specific VOCs or specific groups of VOCs (i.e., aldehydes, methane, aromatics, and methanol); a white square indicates that a predictor was not included in that particular run (runs are listed along the y-axis)*

(White indicates that a predictor was not included in a run, and the VOC species subsets are defined in Figure S5)

[Figure]

5 *S6 – Illustration of selected periods of different relative composition*s of difference VOCs and groups of VOCs *; these are the subsets utilized for the analysis in Section 3.2 of the manuscript and shown above in Figure S5*

[Figure]

*Figure S7: Sensor signal vs. temperature and pollutant concentrations (Section 3.2); the sensor signals posted here are the normalized resistance values, not yet calibrated; the colors represent pollutant concentrations (in panels a and b summed VOCs in ppbC are shown, and in panels c and d methane is shown in ppmV)*

[Figure]

*Figures S8: Regression analysis results, excluding VOC sensor signals (Section 3.2); the data plotted in blue is from the original analysis conducted in Sections 3.1.1 – 3.1.3; the data plotted in purple is the same regression analysis, utilizing Model 1, with the sensor signal excluded leaving only environmental parameters and time as predictors*
(training data before 7/20 & after 8/5, remaining data is testing)

[Figure]

*Figures S9: Bootstrap analysis for sensor set in secondary U-Pod (Section 3.2); this analysis was conducted in the same with, with a major difference being that less data was available for this second U-Pod as a result of power issues*

[Figure]

*Figures S10: Regression analysis results for sensor set in secondary U-Pod; these plots illustrate the results of the regression analysis for the secondary U-Pod as well as which data was available for analysis, given power failures that occurred for this second U-Pod*

10 (training data before 7/20 & after 8/5, remaining data is testing)

[Figure]

[Figure]

[Figure]

[Figure]

**Figures S11: Complete sensor ratio plots (Section 3.3)**; *in all the following plots baseline corrected data from the NATIVE Trailer reference instruments are plotted with colors to indicate where particular VOC sensor ratio fall*

[Figure]

Fig1/Fig2 Ratio
(R/R0, w/
baseline removed)

■ 1.75 < r
■ 1.15 < r < 1.75
■ 0.65 < r < 0.95
■ 0.40 < r < 0.65
■ 0.00 < r < 0.40